behaviour, evolution, ecology

hearing threshold, audiogram, auditory brainstem responses, dynamic range, sensory system evolution, amplitude coding

**Authors for correspondence:**
Ella Z. Lattenkamp
e-mail: ella.lattenkamp@evobio.eu
Mirjam Knörnschild
e-mail: mirjam.knoernschild@mfn.berlin

# Hearing sensitivity and amplitude coding in bats are differentially shaped by echolocation calls and social calls

Ella Z. Lattenkamp[1,2], Martina Nagy[3], Markus Drexl[4], Sonja C. Vernes[2,5], Lutz Wiegrebe[1] and Mirjam Knörnschild[3,6,7]

[1]Department Biology II, Ludwig Maximilians University Munich, Martinsried, Germany
[2]Neurogenetics of Vocal Communication Group, Max Planck Institute for Psycholinguistics, Nijmegen, The Netherlands
[3]Museum für Naturkunde, Leibniz-Institute for Evolution and Biodiversity Science, Berlin, Germany
[4]German Center for Vertigo and Balance Disorders (IFB), Ludwig Maximilians University, Munich, Germany
[5]Donders Institute for Brain, Cognition and Behaviour, Nijmegen, The Netherlands
[6]Animal Behavior Lab, Freie Universität, Berlin, Germany
[7]Smithsonian Tropical Research Institute, Balboa, Ancón, Panama

EZL, 0000-0002-8928-8770; MN, 0000-0002-9768-3930; SCV, 0000-0003-0305-4584; LW, 0000-0002-9289-6187; MK, 0000-0003-0448-9600

Differences in auditory perception between species are influenced by phylogenetic origin and the perceptual challenges imposed by the natural environment, such as detecting prey- or predator-generated sounds and communication signals. Bats are well suited for comparative studies on auditory perception since they predominantly rely on echolocation to perceive the world, while their social calls and most environmental sounds have low frequencies. We tested if hearing sensitivity and stimulus level coding in bats differ between high and low-frequency ranges by measuring auditory brainstem responses (ABRs) of 86 bats belonging to 11 species. In most species, auditory sensitivity was equally good at both high- and low-frequency ranges, while amplitude was more finely coded for higher frequency ranges. Additionally, we conducted a phylogenetic comparative analysis by combining our ABR data with published data on 27 species. Species-specific peaks in hearing sensitivity correlated with peak frequencies of echolocation calls and pup isolation calls, suggesting that changes in hearing sensitivity evolved in response to frequency changes of echolocation and social calls. Overall, our study provides the most comprehensive comparative assessment of bat hearing capacities to date and highlights the evolutionary pressures acting on their sensory perception.

## 1. Introduction

Sensory systems are based on overarching, phylogenetically determined principles but also show species-specific adaptations, thus highlighting the diverse evolutionary pressures acting on sensory perception [1]. In echolocating taxa, such as toothed whales and bats, hearing is the dominant sense used to create a neural representation of the external world by interpreting the returning echoes of self-emitted calls or clicks [2,3]. The perceptual challenges associated with different ecological niches have shaped the design of echolocation calls/clicks considerably [4,5]. It is often argued that these perceptual challenges play a more important role in shaping call design than phylogenetic origin [6]; and this is evident in many examples of convergent evolution of call features from distantly related bat species [7]. However, separating the contributing effects of phylogeny and perception on call design is difficult since some factors, such as beam shape and body size, are shaped by both and can

influence call design considerably as well [8]. In addition to perceiving echolocation calls/clicks and their returning echoes (and a variety of other environmental sounds such as prey- or predator-generated sounds), echolocating taxa also need to perceive social vocalizations, especially since many echolocating species are highly social [9,10]. However, the influence of acoustic communication signals on auditory perception has received little attention in echolocating taxa thus far [11,12].

In bats, echolocation calls are used for orientation, navigation and foraging [13] and, although mostly stereotypic, their acoustic parameters can be adjusted depending on the perceptual task [5]. Even though echolocation calls are not primarily used for communication, they often facilitate it [14]. Social vocalizations of bats consist of calls and songs; they are diverse, flexible, species-specific and vitally important in bat social systems [15]. While social calls are used in a wide range of communicative situations (e.g. parent–offspring interactions or group cohesion [16]), songs are exclusively used for territorial defence or mate attraction [17]. These multi-facetted functions highlight the importance of echolocation calls and social vocalizations for bats, and the strong evolutionary pressure to perceive them accurately.

Hearing in bats is adjusted in a species-specific way to the respective acoustic signal types and the situations in which they are produced [13]. However, hearing in bats should employ common principles to accommodate for the fundamental differences of echolocation calls and social vocalizations. Echolocation calls are generally produced at higher frequency ranges than social vocalizations in a given species [18]. Moreover, echolocation calls need to work over a broad range of distances to detect both near and far objects [3]. Thus, bats need to perceive both their own loud calls and their faint echoes [19] and sometimes even the echolocation calls of other bats [20]. On the other hand, social vocalizations are generally close-range signals that are typically perceived with similar intensities. Notable exceptions are lekking bat species, which attract mates over distances [21]. However, amplitude differences in social calls do not code vital information, such as target distance, size and strength [22,23].

To study bat hearing and investigate common principles across species, comparative data on bat hearing thresholds are crucial. Auditory brainstem responses (ABRs) are acoustically evoked summed electrical potentials that have been established as a fast, objective, and minimally invasive assessment of hearing in different species since the 1970s [24,25]. Furthermore, the ABR growth functions can be related to the dynamic range of hearing, as has been demonstrated for example in bats [26], rats [27] and loudness perception in humans [28], and can thus be used to assess signal level encoding in the auditory pathway. However, only a few authors reported ABRs from bats, despite it being the second largest mammalian taxon with over 1400 extant species [e.g. 29–31]. Previously reported bat ABRs depict not only a sensitivity peak at the frequency range of the species' echolocation calls but often another sensitivity peak at a lower frequency range. This low-frequency sensitivity conceivably allows bats to listen to prey- and predator-generated sounds and may have been retained from the bats' phylogenetic ancestors [3]. Another possibility is that the low-frequency sensitivity evolved in correlation with the frequency content of vital social calls (e.g. pup isolation calls, which are fitness-relevant social signals used by dependent offspring to solicit care).

Correspondingly, one study detected an evolutionary relationship between bat hearing thresholds and the frequency range of a species' echolocation and isolation calls [12]. At present, however, it is unresolved whether bats' low-frequency hearing capacities are mainly influenced by the need to detect prey/predator-generated sounds and/or social vocalizations.

In the present study, we measured tone-pip-evoked ABRs of 11 bat species to investigate the commonalities and differences of their hearing capacities. We also assessed the dynamic range of hearing in high- and low-frequency ranges by calculating the magnitude of the supra-threshold ABRs for each species; this allowed us to identify shared principles of stimulus level coding between the measured species. We hypothesized that the coding of level differences would be more resolved for higher frequency ranges than for lower frequency ranges. Whenever our sample size allowed, we also investigated whether males and females of the same species differed in their hearing thresholds. Moreover, we combined our own ABR data with published information on 27 additional bat species in a phylogenetic comparative analysis to test whether peaks in hearing sensitivity correlated with peak frequencies of echolocation calls or isolation calls. We hypothesized that changes in sensitivity to specific frequencies evolved in response to changes in the peak frequency of echolocation or isolation calls.

## 2. Material and methods

### (a) Animals, auditory brainstem response set-up and recordings, anaesthetics and experimental approval

We measured ABRs from a total of 86 bats from 11 different Neotropical species belonging to six families (electronic supplementary material, table S1). For seven species, no information on hearing thresholds had been available before. All animals were adult and wild caught in Panama, near their roosting sites in the area of Gamboa (9.119925, −79.704512), during March and April 2019. All bats were kept for experimentation in the laboratory of the Smithsonian Tropical Research Institute in Gamboa and were released again within 24 h of their capture. ABRs were measured in a custom-made set-up consisting of a sound-attenuating box, a high-quality speaker connected to an amplifier and an audio interface. Details on the ABR set-up and the calibration process are provided in the supporting information. ABR recordings of anaesthetized bats (see electronic supplementary material, table S1 for the application of anaesthetics) were made with two subdermal electrodes (clipped needles, Sterican brown 0.45–12 mm, B. Braun, Melsungen AG, Melsungen, Germany), which were placed at the caudal midline of the head, close to the brainstem (recording electrode) and at the dorsal midline of the head between the ears (reference electrode) (electronic supplementary material, figure S1b). The ground electrode was either placed on the base of the left ear of the animal or on the wing or tail membrane, if the ears were too small for correct electrode placement. Prior to the positioning of the electrodes, the fur was trimmed with scissors to enable their precise placement. The electrodes were connected via alligator clips to a bioamplifier (BMA-200, CWE Inc., USA), which bandpass filtered the electrical responses (between 100 Hz and 3 kHz) and initially amplified the signal by 60 dB. The signal was AD-converted to digital by the above-mentioned audio interface. The ABR signals were down-sampled by a factor of 20 and each of the 256 recordings (i.e. 256 repetitions of the same frequency-sound level combination) were saved with a final sampling rate of 19.2 kHz. Time-domain averaged ABR

signals for each combination were displayed to the experimenter for quality monitoring during the recordings.

## (b) Stimuli

Recordings of ABRs were done in response to tone-pip stimuli presented in the free field. The tone-pips were sinusoids of 2.5 ms duration (multiplied with an equal-length Hanning window) with carrier frequencies which were evenly spaced between 5 and 120 kHz in eleven steps of approximately half an octave (550 Cent) on a logarithmic frequency axis. These stimuli were generated at a sampling rate of 384 kHz and a digital word length of 24 bit. The pip stimuli were presented 256 times with a 44 Hz repetition rate and in 10 dB increments at sound levels between 0 and 110 dB peak-equivalent sound pressure level (peSPL). The 12 sound levels were again randomized, but all eleven frequencies were presented at one sound level first, before the next sound level was chosen. The order of the tone-pip carrier frequencies within a sound level was also chosen randomly. A custom written MATLAB script (MATLAB, R2018b, MathWorks, Natick, NA, USA) was used to generate the stimuli and coordinate their presentation via the above-mentioned audio interface. Every other stimulus was phase-inverted to cancel out electrical stimulus artefacts picked up by the ABR electrodes after averaging in the time-domain.

The frequency and sound level resolution were chosen for optimal coverage at minimum distress levels for the animals. However, to confirm the robustness of the measured curves, we additionally measured one or two individuals of each species with an increased frequency resolution (electronic supplementary material, figure S2). These additional measurements covered the range from 5 to 120 kHz in 30 steps (logarithmically distributed, resulting in a step size of about 190 Cent) and were instead recorded at only eight sound levels (i.e. 40–110 dB peSPL in 10 dB increments). These higher resolution measurements were not used in the calculation of the mean hearing thresholds as slightly different frequencies were tested.

## (c) Auditory brainstem response data analysis

We evaluated our ABR signals quantitatively and objectively in accordance with previous experiments using the same experimental set-up [32]. The amplitudes of the recorded ABRs were calculated as the root-mean square (RMS) in the time window starting directly after the stimulus presentation and lasting for the duration of the ABR signal (i.e. 1–8 ms after stimulus onset). The RMS of the full signal was used in order to evade unreliability from individual waveform discrimination [33,34] and to increase comparability with the recent literature [32,35]. Bootstrap analyses ($n = 500$; 95% confidence) were performed on the ABR data to statistically verify the presence of an ABR signal [36]. This procedure tests the statistical likelihood that a recorded signal represents random variation in the data rather than a physiological response. To that end, repeated random resampling (with replacement) of the original data was performed and then assessed whether the RMS of the resampled waveform exceeded the original. If 95% of resampled waveforms had a lower RMS than the original waveform, the measurement was considered significant. The lowest sound level evoking a significant ABR signal (at a specific frequency) was conservatively accepted as threshold only when significant ABRs were also obtained for all higher sound levels at the same frequency.

The bootstrap analyses were used to assess the characteristic ABR threshold for each species. Mean thresholds and standard error of the mean (SEM) were calculated per species, omitting measurements for which the algorithm could not determine the threshold. This approach is therefore biased towards lower threshold values since thresholds higher than the highest tested level would not be determined and thus omitted. The hearing threshold or audiogram is thus a calculated value indicating the lowest stimulus level where a statistically significant difference of the ABR signal from background noise occurred. The hearing threshold itself contains no information about the ABR signal amplitude and can represent different ABR signal amplitudes for different frequencies, depending also on the noise floor. The isoresponse lines are not bootstrapped, but instead indicate the species-specific average ABR signal strength (in µV) for each measured frequency-amplitude combination. Isoresponse lines are independent from previous values (e.g. for different amplitudes at the same tested frequency).

To calculate the slope of growth functions at each stimulus frequency, a sigmoidal curve was fitted to each function (using the nlinfit function in MATLAB) and the coefficient of determination ($R^2$) of the fits was assessed, indicating the goodness of fit. A shallow slope suggests a large dynamic range, i.e. stimulus level differences are finely coded at this frequency, while a steep slope indicates a smaller dynamic range. Using ABR growth functions to assess the dynamic range of hearing (i.e. the ratio between the loudest and faintest stimuli that can be detected) has mainly been explored in relation to human loudness perception in the past [28,37,38], but has also been applied in animal models [26,27].

## (d) Phylogenetic comparative analyses

We combined our own ABR data on 11 species with published information on 27 additional bat species to test whether peaks in hearing sensitivity correlate with peak frequencies of echolocation or isolation calls, two highly important call types for bats. In total, our analysis included 38 species from 13 families (echolocation call data: 37 species; isolation call data: 27 species). We thus conducted the phylogenetic comparative analysis with considerably more data than a previous study [12], which included 13 species from six families. For all species, audiograms were available, thus enabling us to estimate peaks in hearing sensitivity. Additionally, for all species the peak frequency of the respective echolocation calls was known; however, in one case, the peak frequency of echolocation calls was outside the range of tested ABR frequencies. For the majority of species, the peak frequency of their isolation calls was also known ($n = 27$).

When studying evolutionary relationships between certain traits it is necessary to account for phylogenetic effects [39]. We thus assembled phylogenetic trees for the echolocation and isolation call datasets using a recent, dated molecular phylogeny of bats [40]. We used the phylogenetic comparative analysis SLOUCH (stochastic linear Ornstein-Uhlenbeck models for comparative hypotheses) to test whether changes in hearing sensitivity in specific frequency regions evolved in response to changes in the peak frequency of echolocation or isolation calls. Ornstein–Uhlenbeck models of trait evolution [41] not only model drift (Brownian motion—part of trait dynamics) but also the rate of adaptation and evolution of a trait (e.g. hearing sensitivity) toward an optimal state (as a linear function of a predictor; e.g. peak frequency of calls); this makes them well suited to test if there was an adaptive correlated evolution between two traits such as hearing sensitivity and call frequencies. We calculated two regression slopes with hearing sensitivity as trait and peak frequency of calls as predictor: (1) the 'optimal regression' slope describes the expected relationship between the trait and the predictor if no constraints on the evolution towards the optimal state existed (i.e. phylogenetic inertia); and (2) the 'evolutionary regression' slope depicts the current relation between the trait and the predictor. To assess how well phylogeny alone explained differences in hearing sensitivity, we also calculated an intercept-only model and contrasted it with the full model that included peak frequency of calls as predictor. We compared both models with AICc scores and assessed

model support with log likelihood values. Statistical tests were conducted in R v. 3.6.0 [42] using the SLOUCH package.

## (e) Extraction of call parameters and hearing sensitivity peaks for phylogenetic comparative analysis

We extracted species-specific call parameters and peaks in hearing thresholds for all 38 species. Acoustic parameters of the bats' calls were extracted from our own recordings or from the literature (electronic supplementary material, table S2). For all species, we reported minimum, maximum and peak frequencies (i.e. frequency with the highest magnitude in the power spectrum) of echolocation calls and isolation calls. Details on the selection of echolocation calls and isolation calls are provided in the supporting information. Moreover, we reported call parameters for contact calls, courtship calls/songs, alert calls and territorial songs whenever this information was available (electronic supplementary material, table S2).

Bat audiograms typically have two sensitivity peaks with a relatively insensitive region in between [3]. However, some bat species, especially those that produce constant frequency echolocation calls, show three frequency regions of increased auditory sensitivity instead of two [43]. As basis for the phylogenetic comparative analyses, we used two sensitivity peaks per species. If three peaks were present, we used the peaks in the lowest and highest frequency regions. This was the case for three species in this study, namely *Rhynchonycteris naso* (figure 1*c*), *Glossophaga soricina* (figure 1*d*) and *Thyroptera tricolor* (figure 1*i*). We determined the frequency of each sensitivity peak by using either the 1 µV isoresponse line (own data) or the audiogram threshold line (data from the literature; electronic supplementary material, table S2). Since isoresponse lines indicate the strength of the ABR signal in µV, using the 1 µV isoresponse line is more conservative than using the threshold line. Hereafter, both are called 'threshold' for simplicity. Following a previous study [12], when two adjacent frequencies with the same threshold constituted the sensitivity peak, we used average values. The majority of audiograms came from anesthetised instead of awake bats (22 versus 16 species). Thus, we probably underestimated some species' sensitivity to high frequencies, because anesthetized bats have a reduced sensitivity to those [3]. Moreover, it is important to note that ABR audiograms are generally about 20 dB less sensitive than behavioural audiograms [29], even though the general threshold shape is comparable.

Our compiled dataset included six species of bats relying on prey-generated sounds during foraging, five species using 'glints' in the returning echoes to detect fluttering insects and 27 species not relying on prey-generated sounds to detect prey (electronic supplementary material, table S3). This enabled us to assess if the need to perceive social call is influencing the low-frequency hearing capacities of bat species that are not relying on prey-generated sounds for foraging.

## 3. Results

We measured ABRs from 86 adult bats belonging to eleven species (*Saccopteryx bilineata, Saccopteryx leptura, Rhynchonycteris naso, Glossophaga soricina, Carollia perspicillata, Desmodus rotundus, Phyllostomus hastatus, Pteronotus parnellii, Thyroptera tricolor, Myotis nigricans* and *Molossus molossus*) from six families (Emballonuridae, Phyllostomidae, Mormoopidae, Thyropteridae, Vespertilionidae and Molossidae). For four species (*S. bilineata, G. soricina, D. rotundus* and *T. tricolor*), we measured at least three individuals per sex; measurements for the other species were either male- or female-biased (electronic supplementary material, table S1).

## (a) Hearing thresholds in relation to echolocation calls and social vocalizations

The tone-pip-evoked ABR thresholds showed a general tub-shaped trend and a large overlap between the different species (figure 1; electronic supplementary material, figure S3). For all measured species, hearing sensitivity steadily increased between 5–15 kHz and showed a general decrease above 50–60 kHz (figure 1; electronic supplementary material, figure S3). The species differed in the slopes and extrema of their ABR curves and showed between 1 and 3 sensitivity peaks in their hearing thresholds (electronic supplementary material, figure S3). The different isoresponse lines of the contour plots (figure 1) indicated the strength of the ABR signal in µV. Peaks in the mean threshold and isoresponse lines corresponded to echolocation frequency ranges for all species (figure 1). This relation could not be assessed for one species, which echolocates at very high frequencies (*T. tricolor*; 147 kHz; electronic supplementary material, table S2) that exceeded the frequency range in which we measured ABRs. Most species also showed peaks in the mean threshold and isoresponse lines in the frequency range of their social vocalizations. For instance, in *S. bilineata* (figure 1*a*), the two most prominent peaks in the first isoresponse line correspond perfectly to the frequency range of echolocation calls (44.3–48.9 kHz; electronic supplementary material, table S2) on the one hand and the frequency range where important social vocalizations overlap on the other hand (isolation calls, territorial and courtship songs; 17.7–22.1 kHz; electronic supplementary material, table S2). Also, *T. tricolor* (figure 1*i*), the species for which we could not measure ABR thresholds in the frequency range of their echolocation calls, showed a very pronounced peak in the frequency range of important social vocalizations (isolation calls and contact calls (49.2–78.4 kHz; electronic supplementary material, table S2). Sonograms of representative echolocation calls and isolation calls are provided in electronic supplementary material, figure S4.

## (b) Dynamic range of hearing in high- and low-frequency ranges

The distances between the isoresponse lines can be expressed as ABR growth functions for each frequency (figure 2). The shallower the slope of the growth function, the larger is the dynamic range of ABRs, an indication of more detailed coding of the bat's response to different stimulus levels at a given frequency. For higher frequency ranges, corresponding to echolocation calls, the ABR growth functions were significantly shallower than for lower frequency ranges, corresponding to social vocalizations (Wilcoxon signed-rank test; $Z = -2.1$, exact $p = 0.036$, $n = 10$; *T. tricolor* was not included since their echolocation calls exceeded the frequency range of our ABR measurements), and ABR amplitude saturation is rarely encountered in the tested level range (up to 110 dB SPL). This finding indicates that reliable loudness coding is more important for echolocation frequency ranges, where loudness is a cue for echoacoustic target distance and size. The steepest growth functions for all tested species were found between 5 and 20 kHz. For some species, namely *S. leptura, G. soricina* and *D. rotundus*, the slopes increased again towards higher frequencies (figure 2*b,d,f*).

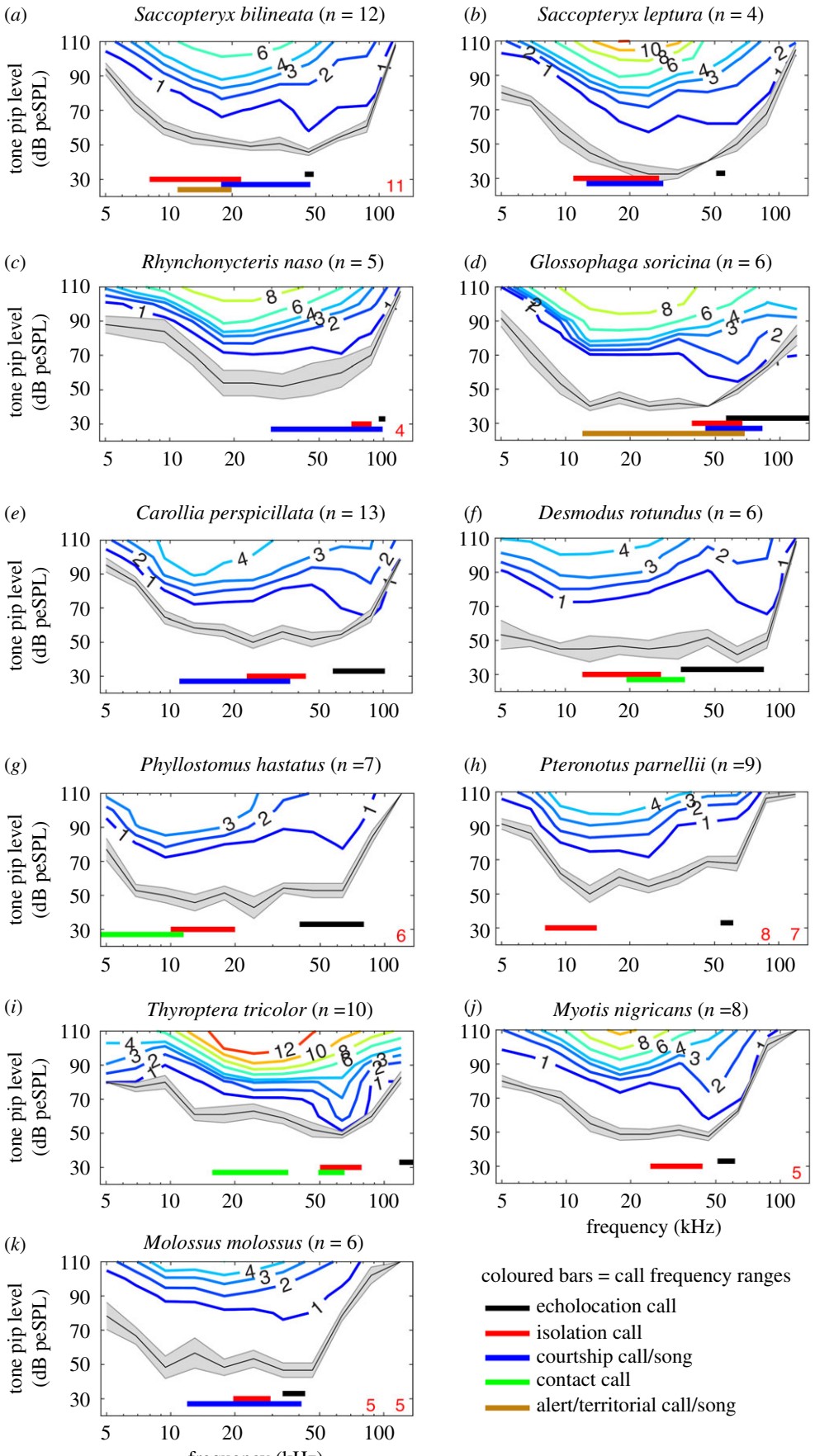

**Figure 1.** Species-specific mean ABR thresholds calculated via bootstrap analysis for 11 species from six families: Emballonuridae (a–c), Phyllostomidae (d–g), Mormoopidae (h), Thyropteridae (i), Vespertilionidae (j) and Molossidae (k). The mean ABR threshold per species is depicted (black line; shading represents SEM). The isoresponse lines represent the strength of the ABR signal (colours and numbers indicate µV response strength). The number of animals measured per species is given in the plot titles (n). Vertical bars below the ABR thresholds indicate bandwidth of five different call types: echolocation calls (black), isolation calls (red), courtship calls (blue), contact calls (green) and territorial/alert calls (brown). For details, see electronic supplementary material, table S2. Red numbers on the bottom of the panels indicate the number of animals used for the calculation of mean and SEM (if less than total sample size). The numbers are positioned above the frequency for which mean values were determined.

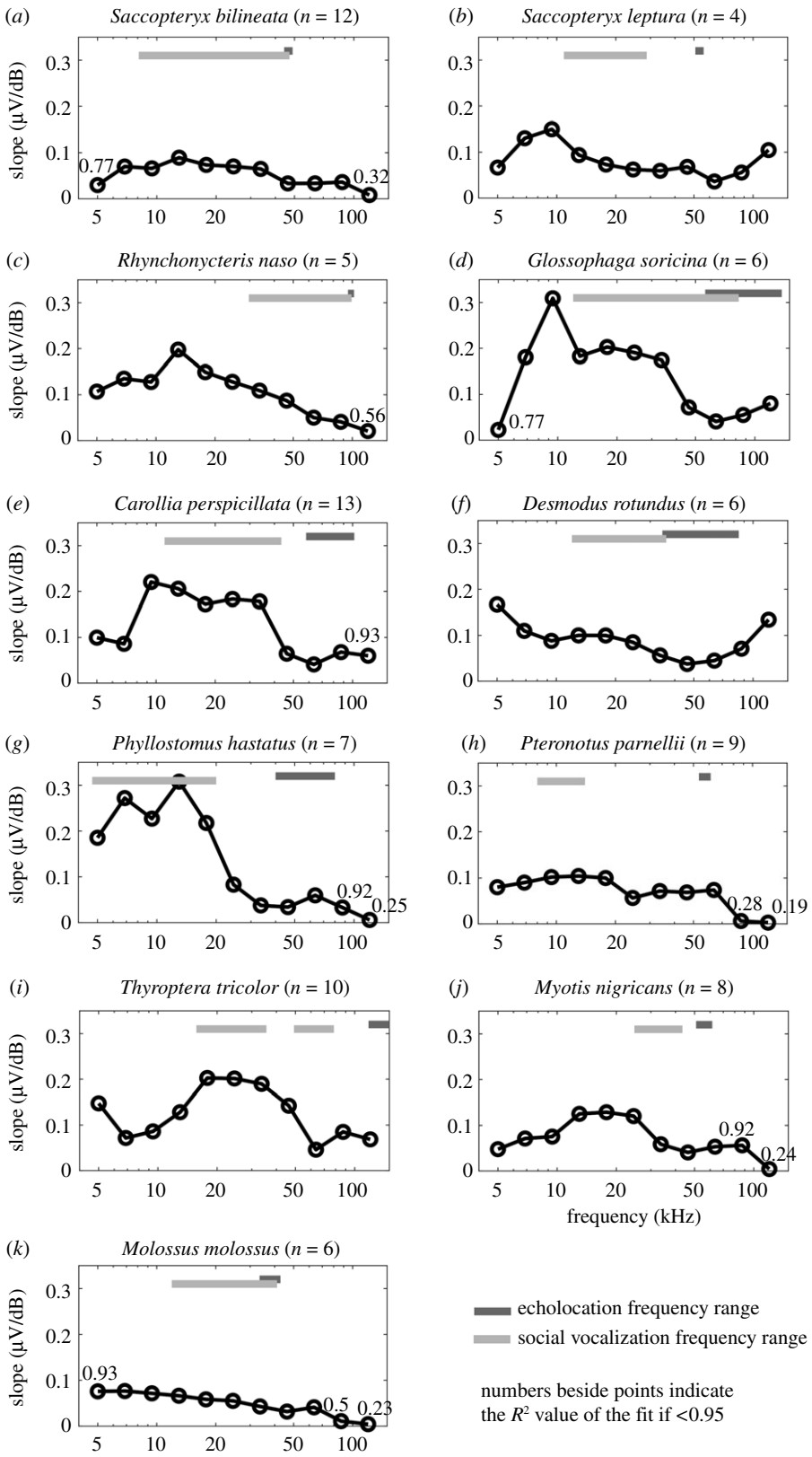

**Figure 2.** Species-specific slopes of ABRs' growth functions for each frequency. The $R^2$ quality measures of the fits are given above each frequency (if less than 0.95). A shallow slope corresponds to a large dynamic range of stimulus amplitude encoding (i.e. the more finely coded is the bat's response to different stimulus amplitudes at a given frequency); a steep slope corresponds to a small dynamic range. Dark grey bars indicate the frequency ranges of echolocation calls, while the light grey bars indicate the frequency ranges of social vocalizations.

## (c) Sex differences in hearing sensitivity

We also compared ABR thresholds of males and females whenever our sample size allowed it (electronic supplementary material, table S1; *S. bilineata, G. soricina, D. rotundus* and *T. tricolor*). In two species, *S. bilineata* and *D. rotundus*, females' ABR thresholds were more sensitive than males' in a frequency range corresponding to pup isolation calls; ABR thresholds of males and females did not show an overlap of the SEM in this frequency range (electronic supplementary material, figure S5a,b). However, in two other

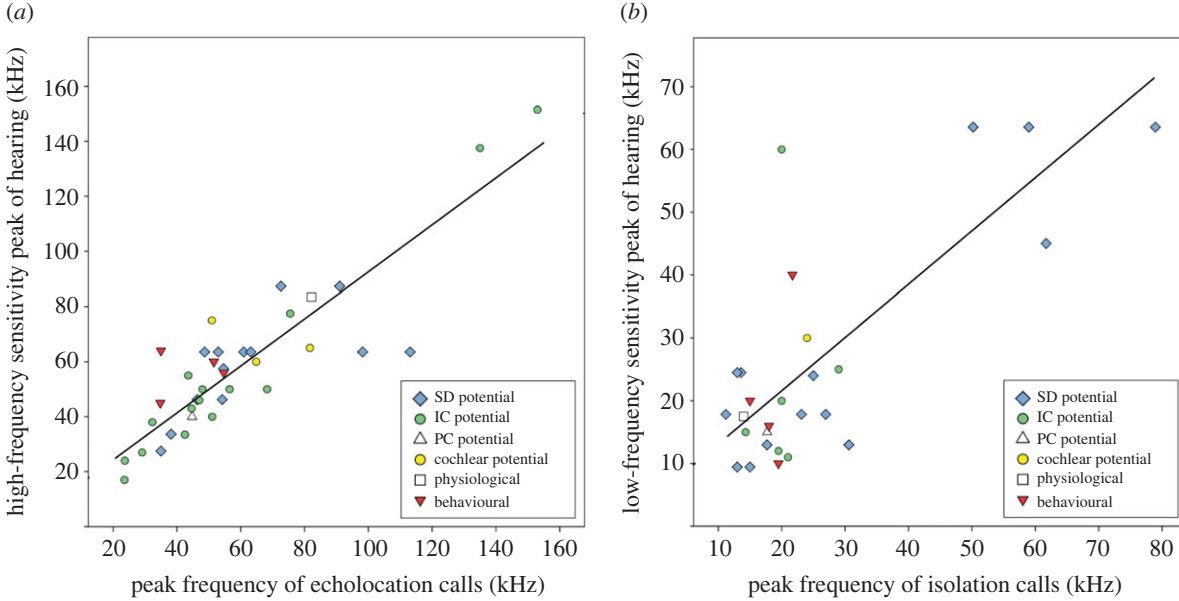

**Figure 3.** Positive relation between hearing sensitivity (in higher and lower frequency ranges) and the peak frequency of echolocation calls (a) and isolation calls (b). Graphs depict the original data with the fitted optimal regression slopes from phylogenetically corrected analyses (OU models of trait evolution). Details on OU models as well as data on call frequencies and sensitivity peaks for all species in our phylogenetic comparative analysis are provided in the supplementary material (electronic supplementary material, tables S3–S4, figure S6). Different symbols depict electrophysiological (SD, IC, PC and cochlear potential), physiological (change in heart rate) and behavioural (discrimination paradigm) ways in which hearing thresholds were determined. Electrophysiological data were collected from extra-cellular recordings subdermally (SD) at the brainstem, at the inferior colliculus (IC) and posterior colliculus (PC) in the auditory midbrain, and at the round window of the cochlea (cochlear potential). (Online version in colour.)

species, *G. soricina* and *T. tricolor*, the ABR thresholds' overlap of SEM was extensive, especially in the frequency range of pup isolation calls (electronic supplementary material, figure S5c,d).

### (d) Peaks in hearing sensitivity correlate with echolocation and isolation call frequencies

Our phylogenetic comparative analyses, based on Ornstein-Uhlenbeck models of trait evolution, clearly demonstrated a positive relationship between hearing sensitivity in higher frequency ranges and the peak frequency of echolocation calls (optimal regression slope = 0.824, 95%-CI = 0.969–0.679, $R^2 = 0.767$, $n = 37$; figure 3a). We also detected a positive relationship between hearing sensitivity in lower frequency ranges and the peak frequency of isolation calls (optimal regression slope = 0.799, 95%-CI = 1.056–0.542, $R^2 = 0.577$, $n = 27$; figure 3b). Details on the models are provided in electronic supplementary material, table S4 and figure S6. Overall, our data indicate that an evolutionary change in the peak frequency of echolocation calls and isolation calls was associated with corresponding changes in hearing sensitivity in the higher and lower frequency ranges.

The observed correlated evolution between low-frequency sensitivity peaks and the frequency range of pup isolation calls is clearly present in species not relying on prey-generated sounds to detect prey. Out of 28 bat species not relying on prey-generated sounds, 19 species had sensitivity peaks in the low-frequency range of their pups' isolation calls. Five additional species also had low-frequency sensitivity peaks, but pup isolation calls of these species are undescribed. Only four species not relying on prey-generated sounds had no low-frequency sensitivity peaks at all (electronic supplementary material, table S3).

## 4. Discussion

In this study, we measured ABRs of 86 bats from 11 species and compiled data on 27 additional bat species from the literature. For 11 species, we not only assessed hearing sensitivity but also stimulus level coding in high and low-frequency ranges (by calculating the ABR growth function). This provides the most comprehensive comparative assessment of the hearing capacity of bats to date. We evaluated our ABR signals quantitatively and objectively by calculating the signal's root-mean-square amplitude and determining its significance via bootstrapping (following [32]), thus facilitating further comparisons in the future.

### (a) Auditory brainstem responses are useful tools for cross-species analyses of hearing capacities

In all species we measured, the ABR thresholds showed a general tub shape, even though species differed in the frequency of their respective sensitivity peaks. Our findings show that hearing thresholds are characteristic despite the existing overlap between species (electronic supplementary material, figure S3). The small variation between hearing thresholds within species (shown by the small SEM in figure 1) and the comparably large variation between species (electronic supplementary material, figure S3) indicates species-specific adaptation to their ecological or evolutionary niches. For example, *D. rotundus* shows an unusually high hearing sensitivity in low-frequency ranges (less than 10 kHz; figure 1f), which was previously shown to support prey-generated-noise detection in these sanguivorous bats [44]. The thresholds measured in this study are very comparable with previously published audiograms (for details, see electronic supplementary material). ABRs are a useful tool to assess response strength to a large, consistent and

reproducible parameter space of auditory stimuli in a large number of animals. We argue that this consistency in tested parameters is optimal for a cross-species comparative approach. Moreover, our approach also allowed us to analyse frequency-specific ABR growth functions, which can be used to investigate the dynamic range of hearing and compare sensory capacities of different species.

## (b) Amplitude coding in bats follows a general principle

Our findings show that bats have a significantly larger dynamic range of hearing in the high-frequency range of echolocation calls than in the low-frequency range of social vocalizations. This means that the amplitude of high-frequency vocalizations is more finely coded in the auditory pathway than for low-frequency vocalizations. Echolocation calls are typically very loud (call intensities can reach up to 140 dB SPL at a distance of 0.1 m from the bat's mouth [45]), but at the same time, the returning echoes can be quite faint [19]. Moreover, the sound level of echolocation calls is dynamically adjusted based on the habitat, the distance and the target strength of the ensonified object [22]. Also, echo amplitude is an important predictor of target size [23]. Thus, amplitude differences of high-frequency sounds carry vital information for bats and the large dynamic range of hearing at these frequencies may be an adaptation to cope with these intensity differences. Social vocalizations, on the other hand, are often emitted and perceived at similar intensities as they are typically used for communication in relative proximity. Our findings show that although the frequency range of echolocation calls and social vocalizations is species-specific, all eleven measured bat species show the same general principle of level coding: a shallow ABR growth function (i.e. large dynamic range of hearing) in the frequency range of their echolocation calls and a steep ABR growth function (i.e. small dynamic range of hearing) in the frequency range of their social vocalizations. It seems straightforward to assume that this is a shared principle between all laryngeally echolocating bats. It would be interesting to study this phenomenon in bat species that do not rely as strongly on the detection of minute level differences in a specific frequency range, such as pteropodid bats.

## (c) In some bat species, females are more sensitive than males in low-frequency ranges

In bats, sexual dimorphism in the production or use of acoustic signals is relatively common [14,16], while evidence for sexual dimorphism in hearing sensitivity has not been published thus far. For two species (*S. bilineata* and *D. rotundus*), we found differences in ABR thresholds between males and females in the frequency range of their social vocalizations, especially in the range of pup isolation calls, but not in their echolocation calls. These results are in concordance with an earlier study on *P. hastatus*, which demonstrated that females are especially sensitive to the frequency range of pup isolation calls via behavioural audiograms [46]. For two other species (*G. soricina* and *T. tricolor*), we detected no differences in hearing sensitivity between the sexes. These results need to be interpreted with care as the sample size for each sex is limited. Therefore, at present, we can only speculate whether sexual

dimorphism in hearing sensitivity is commonly expressed in bats and which function it might serve.

## (d) Hearing sensitivity peaks in bats evolved in correlation with the species-specific frequency ranges of echolocation calls and social vocalizations

Our analyses showed that auditory perception in bats is not only shaped and constrained by their faculty of echolocation [3,7] but also by the vocalizations that bats use for social communication. Our results are in accordance with a previous study [12], which also concluded that bat hearing thresholds evolved in correlation with the frequency range of a species's echolocation calls and isolation calls. We extended this former study considerably by including more species in the comparative analysis and by providing more detailed acoustic data on the respective vocalization types involved. Moreover, we confirmed that a low-frequency sensitivity peak is commonly found in species which are not relying on prey-generated sounds to detect prey, thus highlighting the role of social communication in shaping auditory perception. It is conceivable that in other echolocating taxa, such as toothed whales, tenrecs, oilbirds and swiftlets, hearing sensitivity also showed a correlated evolution with social vocalizations [47–49] but, to our knowledge, this phenomenon has not previously been explored with phylogenetic comparative analyses.

## 5. Conclusion

Our large-scale comparison of hearing capacities in bats not only allowed us to investigate commonalities and differences between species but also to identify a species-independent, overarching principle for the perception of different signal types. Amplitude is more finely encoded in the high-frequency range of echolocation calls than in the low-frequency range of social calls, while auditory sensitivity is equally good at both high- and low-frequency ranges. Moreover, we found tentative evidence that, at least in some species, females have higher hearing sensitivity than males in the low-frequency range. In addition, our phylogenetic comparative analysis revealed that the observed hearing sensitivity peaks align with the dominant frequency ranges of echolocation calls and social vocalizations. Overall, our analyses show that the evolution of bat auditory perception is shaped by the pressure to accurately perceive both echolocation and social calls, and thus provides an incentive for the in-depth investigation of evolutionary pressures acting on the sensory perception of other echolocation taxa.

Ethics. The research was conducted in accordance with the Panamanian government (MiAmbiente permit no. SE/A-5-19) and the regulations of the Smithsonian Tropical Research Institute (STRI ACUC protocol 2019-0302-2022).

Data accessibility. Data available from the Dryad Digital Repository: https://doi.org/10.5061/dryad.5hqbzkh4k [50].

Authors' contributions. L.W., M.K. and E.Z.L. conceived the experiment. L.W., M.D. and E.Z.L. designed and tested the set-up. E.Z.L., M.K. and M.N. collected the data. E.Z.L., M.D. and L.W. analysed the ABR data. M.K. and M.N. analysed the acoustic data and conducted the phylogenetic comparative analyses. E.Z.L. and M.K. wrote the manuscript. All authors contributed to the final version of the manuscript.

Competing interests. The authors have no competing interests to declare.

Funding. This study was financed by a Heisenberg Fellowship (DFG KN935 5-1) awarded to M.K. and a Human Frontier Science Program

Research grant no. (RGP0058/2016) awarded to L.W. and S.C.V. S.C.V. was also funded by the Max Planck Society. M.D. received funding from a grant of the German Ministry of Education and Research (BMBF) to the German Centre for Vertigo and Balance Disorders (01EO1401). E.Z.L. was funded by a short-term stipend of the German Academic Exchange Program (DAAD) (no. 57438025) and a travelling fellowship of the Company of Biologists (JEBTF18113). Moreover, the research leading to these results has received funding from the European Research Council under the European Union's Horizon 2020 Programme (2014–2020)/ERC GA 804352.

Acknowledgements. The authors thank the Smithsonian Tropical Research Institute and especially Rachel Page for providing excellent research conditions in Panama. We further thank Constance Scharff, Arturo Zychlinsky, Lorenz Knackstedt, Ine Alvarez van Tussenbroek, May Dixon and Inga Geipel for their help and support during bat capture and measurement. We are grateful to Gloriana Chaverri, Inga Geipel and Gloria Gessinger for providing sound recordings for our analyses. Lastly, we thank the workshop of the Ludwig Maximilians University in Munich for their help building the mobile ABR set-up.

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
