## [Reviewer comments · Proceedings of the Royal Society B: Biological Sciences]

Review History

RSPB-2020-1170.R0 (Original submission)

Review form: Reviewer 1

Recommendation

Reject – article is scientifically unsound

Scientific importance: Is the manuscript an original and important contribution to its field?

Marginal

General interest: Is the paper of sufficient general interest?

Acceptable

Quality of the paper: Is the overall quality of the paper suitable?

Marginal

Is the length of the paper justified?

Yes

Should the paper be seen by a specialist statistical reviewer?

No

Do you have any concerns about statistical analyses in this paper? If so, please specify them explicitly in your report.

No

It is a condition of publication that authors make their supporting data, code and materials available - either as supplementary material or hosted in an external repository. Please rate, if applicable, the supporting data on the following criteria.

Is it accessible?

Yes

Is it clear?

Yes

Is it adequate?

Yes

Do you have any ethical concerns with this paper?

No

Comments to the Author

The manuscript uses a combination of new ABR recordings from 11 species of bats with a meta-analyses of published data to explore two different but related questions: The first asks whether or not the auditory sensitivity range of a bat is influenced by both echolocation and social calls, and secondly whether the dynamic range of coding amplitude is different for high (echolocation) versus low (social call) frequency sounds. The introduction also raised a question about whether low-frequency hearing is influenced by the need to detect social calls or rather prey-generated noises (line 84). As it relates to the first question, the paper seems to come to very nearly the same conclusion of a similar paper (citation 11, Bohn et al., 2006) that also used meta-analyses to estimate the relative influence of social calls and echolocation on evolution of hearing thresholds. It is not made very clear what new conclusions are drawn from this newer analysis. The second question, that of dynamic range coding, has not been explored comparatively, and is a new and noteworthy idea. However, the paper makes it very difficult to discern what is actually meant by this, and it took me several reads through to understand how this was related to echolocation and communication. I think the average reader will struggle getting through the abstract.

While ABRs are widely used as a comparative tool for studying hearing thresholds between species, I am unaware of this tool being used to evaluate dynamic range, and after reading the paper I am not convinced that ABRs are a reliable way to do this. ABRs are complex summated potentials that may not be linearly or directly related to signal amplitude, because they reflect both an increase in individual neuron activity and also the total number of neurons excited. This is problematic if the auditory system has an acoustic fovea composed of a disproportionately larger number of neurons processing a particular important frequency range, and may not be directly related to minimum thresholds or amplitude coding per se at every frequency. So, at the very least the manuscript needs to build support for using ABRs to answer this question. Also, the manuscript mentions identifying “overarching principles or amplitude coding”, but doesn’t provide any background into what this means for mammals, vertebrates, or auditory systems overall. More generally, the rationale for the hypothesis that social calls might require less dynamic range than echolocation calls is not convincing.

Specific comments:

Line 160-162: Is this a new way of calculating dynamic range, or is this an established method in the field? Has it been done using ABR data before?

Line 242 and Figure 1. "The two most prominent peaks in the first isoresponse line..." Why do the peaks only show up in the first line, but not in the mean ABR threshold line or the second line? Similarly, why isn't there a sensitivity peak at 60 kHz for the *Pteronotus parnellii* graph? Isn't this species also well known for using multiple harmonics, at 30 and 90, and 120 kHz? For *P. parnellii* the first isoresponse line indicates rather poor sensitivity in the region where they should have the greatest sensitivity, which may reflect the limitations of using ABRs in this way. A quick review of literature shows that some of these ABR graphs are not very consistent with the literature (see for example Grinnell 1970, JCP 68:117-153). The text should address these discrepancies.

Figure 2. Superficially, this should be a simple transformation, and yet the data in figure 2 raise several questions. First, if the ABR threshold is very high, then this would automatically constrain the growth function, no? Line 161-162 says that a shallow slope represents a large dynamic range, and vice versa. Choosing *Molossus molossus* as an example, it is confusing why in fig 2 this animals dynamic range is highest above 50 kHz, and yet fig 1 shows that it can barely even detect frequencies about 50 kHz. The slope for *P. parnellii* similarly raises questions, because it seems unlikely that for this animal the highest dynamic range would not be around its auditory fovea at 60 kHz.

Figure 3: The authors correctly note that ABR thresholds are not as low as those obtained with behavioral audiograms. ABRs vary considerably between labs, experimental conditions, and hardware being used. I will simply state that an n=3 per sex is probably insufficient to determine a sex difference in ABR thresholds. This point is only loosely related to the rest of the manuscript is not essential to the question of amplitude coding. Behavioral audiograms showed that male vampire bats displayed auditory thresholds 40-60 dB less than those shown here (Heffner et al 2013, Hearing Res., 296: 42-50), so it is hard to believe that male vampire bats are truly 20-30 dB less sensitive than females, especially across their entire hearing range. Recommend removing this section from manuscript and focusing on amplitude coding.

Review form: Reviewer 2

Recommendation

Accept with minor revision (please list in comments)

Scientific importance: Is the manuscript an original and important contribution to its field?

Excellent

General interest: Is the paper of sufficient general interest?

Good

Quality of the paper: Is the overall quality of the paper suitable?

Excellent

Is the length of the paper justified?

No

Should the paper be seen by a specialist statistical reviewer?

No

Do you have any concerns about statistical analyses in this paper? If so, please specify them explicitly in your report.

No

It is a condition of publication that authors make their supporting data, code and materials available - either as supplementary material or hosted in an external repository. Please rate, if applicable, the supporting data on the following criteria.

Is it accessible?

Yes

Is it clear?

Yes

Is it adequate?

Yes

Do you have any ethical concerns with this paper?

No

Comments to the Author

One goal of this interesting manuscript (ms) was to measure the evoked auditory brainstem response (ABR) in 11 Neotropical bat species in response to both acoustic clicks and pure tones. Tonal ABRs were used to construct threshold tuning curves (audiograms), and suprathreshold ABR waveforms were used to examine amplitude coding within a frequency (i.e. the growth of the ABR waveform at different frequency-amplitude combinations), with a special focus comparing the low and high frequency hearing ranges across species: these ranges correspond to pup isolation calls and adult echolocation calls, respectively. The ms also looked at sex differences in ABR thresholds in four bat species where at least three individuals were measured, and found evidence that females had lower ABR thresholds at lower frequencies (where pups emit isolation calls) in two species but not in two others. The ms conducted a further comparative analysis by combining their ABR data with audiogram/hearing information obtained from the literature for an additional 27 bat species where the animals were tested with a variety of auditory electrophysiological recording techniques. The comparative analysis found that species-specific peaks in auditory sensitivity correlated reasonably well with the peak spectral frequencies of echolocation calls and pup isolation calls for that species. The ms concludes by suggesting that changes in hearing sensitivity evolved with changes in the frequency content of echolocation and social calls, and highlights the importance of social communication as an evolutionary pressure acting on auditory perception in bats.

Overall, I enjoyed the science in this ms. The results were noteworthy and important. I agree that this is one of the most comprehensive comparative assessments of the hearing capabilities of bats. Different species appear to have some differences in the overall shape of their mean (and 95% confidence intervals) clicked-evoked ABR waveforms presented at a very loud level (100 dB SPL); however, a between-species comparison was not conducted on these data. The overall shape of the tone-evoked ABR audiograms were somewhat similar but also showed some differences across species, but again there was no quantitative comparison of absolute thresholds or audiogram shape within or between species. There were some differences in the shape and strength of the averaged ABR signals at each frequency-amplitude combination, observed as different distances between bootstrapped iso-response contour lines within an audiogram and plotted as ABR growth functions. The ms reports that ABR growth functions at high frequencies (corresponding to the spectral range of echolocation calls) had shallower slopes compared to the generally steeper sloped ABR growth functions measured at lower frequencies (corresponding to the spectral range of pup isolation calls).

I appreciate the great effort that went into collecting and analyzing these data, which in general are of very high quality. The authors should be congratulated. One thing I did not enjoy was the large amount of information sequestered into the Electronic Supplementary Material. In general, I am never a fan of papers where there is more information in the Supplementary Material than in the paper itself. This seems to defeat the purpose of a stand-alone article. If the journal will allow, I encourage the authors to move all of the relevant and important details of ABR recording, signal processing, and data analysis into the ms proper because the majority of the ms relies on these data. There were a few places where the text was unclear or confusing and I have highlighted them (see Specific Comments) with a goal to improve the flow and overall readability of the ms.

Lastly, all of the figures and tables are of high quality. In my opinion, Figures 1 and 2 are too small and/or over complicated and this takes away from the important information presented in them.

SPECIFIC COMMENTS

Introduction

1. (lines 38-46) It is difficult to separate the relative importance of the role of perceptual challenges associated with different echolocation niches and species phylogenetic origin. Other important factors related to the evolution (phylogeny) and perception (physics and neurobiology) of the spectral design of echolocation and social calls are beam shape of the signal and animal body size (e.g. Thiagavel et al. 2017, 2018; Barclay and Brigham 1991).

2. (lines 63-71) I agree that social calls are often emitted in close-proximity to conspecifics and heterospecifics. But comparing the functional ranges of these two general classes of vocalizations is difficult. If social signals are generally lower in frequency than echolocation calls, then they will attenuate more slowly, propagate further, and this could extend their functional range in comparison to that of echolocation signals (5-20 m). For example, some lekking bats emit vocalizations to attract mates, likely from distances further away than the functional range of echolocation. And except for a few species (e.g. *Desmodus rotundus*, *Diaemus youngi*, and *Diphylla ecaudata*), it remains unknown if bats emit social calls to maintain contact with nearby conspecifics and/or to guide their offspring during migrations or to new roost locations (e.g. Ripperger et al. *Biology Letters* 15(2): 20180884).

3. (lines 68-71) Minor point. This is an interesting idea, but I encourage the authors to consider editing this sentence so it does not presuppose that “bats do not need to perceive the echoes of their own social vocalizations.” I am unaware of evidence that bats use/do not use echoes of social vocalizations to obtain information (but I also know that bats never cease to amaze bat researchers).

4. (lines 83-85) A non-mutually exclusive alternative possibility to why bats have low-frequency hearing (in addition to listening for prey-generated sounds or pup isolation calls) is that sensitivity to low frequencies was crucial and selected for in the earliest mammals for general alertness and detecting predator-generated sounds (e.g. see Fig. 1 in Fullard 1988 *Experientia* 44, 423-428). Perhaps this is what was meant by saying this “may be a remnant from the bat’s phylogenetic ancestors” but the word remnant is tricky because it can suggest that an ability is a relic rather than still functionally useful and adaptive.

Methods

5. (lines 101-109) All of the data were collected in Panama but the ms does not say where... indoors in a lab/building or in the field (i.e. more details please)? The ms proper has more information and space devoted to the phylogenetic analysis than the auditory analysis. Because most of the Results rely on the ABR data, I think the methodological information about ABR electrophysiology and recording procedures should be moved into the main body of the ms

because it is too important to be relegated to Supplementary Material.

6. (lines 112-115) The ms presents clicked-evoked ABRs measured at a high sound pressure level but does not say why click ABRs were collected and their value to the results and data analysis? Figure S3 shows that the mean click-evoked ABR waveforms at 100 dB SPL were quite variable within and between bat species, but there was no further analysis demonstrating whether this variation was meaningful/indicative or predictive of the species. Moreover, there are many sources of variation that could influence the strength and appearance of an ABR waveform (e.g. differences in anesthesia state, electrode placement, skull thickness, electrical noise at time of recording, etc.). Clicked-evoked thresholds were ~60 dB peSPL. Is this because the clicks were shorter in duration and had less energy than the 2.5 ms pure tones? I encourage the authors to add a bit more clarifying text describing the value of collecting and presenting the click-evoked data in the ms given almost all of the ms focuses on the tone-evoked data.

7. (lines 154-157) This analysis assesses the growth in the suprathreshold ABR response with increasing stimulus level but does not compare ABR thresholds within or between a species (except later for a sex difference comparison in four species).

8. (lines 157-162) I was a bit confused by the description of the slope calculation. I assume that the plotted functions were the amplitudes of the evoked ABR signals at each SPL, and that the slope has units of $\mu\text{V}/\text{dB}$ (the units are missing in Fig. 2). Later (lines 160-161), the text suggests the dynamic range at a given frequency is the ratio of the loudest and faintest stimuli that can be detected. It would be helpful and less confusing to the reader if the definitions and calculations of the slope and the dynamic range of hearing were explicit.

Results

9. (lines 230-240) The Abstract says “ABR thresholds differed between species but showed a general tub-shaped trend” ... and later (lines 25-26) ... “In most species, auditory sensitivity was equally good at both high and low frequency ranges.” I mention this text from the Abstract here because the Results section did not compare ABR thresholds between the low and high frequency ranges within and/or between species. There was a comparison of ABR growth functions between the low (pup isolation calls) and high (echolocation calls) frequency ranges, and between males and females in 4 species, but not a comparison of thresholds. I’m not suggesting that the authors conduct a threshold comparison, but they should clarify the text of the Abstract and Results to make it clear that a comparative analysis of threshold tuning was not conducted.

10. (lines 235-236) Minor point. To help your readers, please specify the audiograms in Fig. 1 that have three (3) sensitivity peaks and whether they exist in the threshold tuning curves or in the bootstrapped iso-response lines.

11. (lines 252-254) The ms here could be clarified so that readers do not think the distance between iso-response lines is the ABR growth function. Also, please see point #8 on clarifying the calculation and slope units. I was confused by the word “flexible” to describe the two classes of ABR growth functions. How are ABR growth functions with shallower slopes more “flexible” than ABR growth functions with steeper slopes? My understanding is that growth functions with shallower slopes can encode changes in stimulus SPL more finely compared to growth functions with steeper slopes.

Discussion

12. (lines 300-304) I’m not sure I understand the point that is being made. Behavioral testing often uses a successive approximation procedure to measure thresholds, but I don’t understand why measuring ABRs is better for consistently testing responses to the same stimuli compared to using behavior to consistently test responses to the same stimuli? For example, both techniques can be consistently applied, both have multiple sources of variation, both can adapt/habituate,

and both can fail. Perhaps I have misunderstood what is trying to be communicated?

13. (lines 351-352) Minor point. The results of Figure 4b were based on one type of social vocalization—pup isolation calls—so I suggest a slightly more conservative rewording of the second clause of this sentence so that it focuses on pup isolation calls that facilitate care and mother-offspring communication.
14. (lines 367-369) See point #11 about “flexibly” encoding SPL. What is meant by “auditory sensitivity”? Is this just absolute threshold? Figure 1 suggests that different species may not be equally sensitive in the low and high frequency ranges.
15. (Figure 1) This figure is critical, but the size of each plot is too small, and the panels seem overly complex. The numbers on the iso-response contour line are very small for my eyes, as are the red numbers above the x-axes. There is a wealth of color on this plot, including for other types of social vocalizations that were not the focus of the ms (i.e. is it necessary to show the bandwidths of other types of social calls when the ms is about echolocation versus pup isolation calls? Also, it is hard to tell the difference between red and pink at the small size)? I encourage the authors to make Figure 1 a full-page width, with taller and wider panels and larger font sizes.
16. (Figure 2) This figure plots the slope of the ABR growth function at each frequency. The units on the y-axis are not reported (i.e. $\mu\text{Vs}/\text{dB}$?). Again, each panel is too small. The R2 numbers are nearly impossible to read without magnification. The panels should be taller and wider, so the figure is a full-page width in the journal. The authors might consider standardizing the location of the bars representing the bandwidths of pup isolation calls and echolocation calls across all figures if they thought this would help improve readability.
17. (Figure 4) This is another very important figure. The caption for the legend says “Different symbols depict ways in which hearing thresholds were determined. SC: subcutaneous {potential} (ABRs), IC: inferior colliculus {potential} (electrophysiological recordings), PC = posterior colliculus {potential} (electrophysiological recordings).” The legend is a bit confusing because it is a mixture of electrophysiological recording techniques and anatomical locations. All three symbols (SC, IC, PC) refer to recording electrophysiological potentials but IC and PC reference to central auditory nuclei. The location of PC is undefined (i.e. does PC refer to the auditory thalamus and/or cortex or the auditory midbrain and/or brainstem?). Were all of the data collected with extracellular recording or were some from intracellular recording? The caption does not mention filled triangles and behavioral thresholds.
18. (Figure S2) Please specify if any of the data recorded with the higher frequency resolution technique were used in the final analysis (e.g. to measure ABR thresholds or to obtain ABR growth functions).

Decision letter (RSPB-2020-1170.R0)

29-Jun-2020

Dear Dr Knörnschild:

I am writing to inform you that your manuscript RSPB-2020-1170 entitled "Hearing sensitivity and amplitude coding in bats are differentially shaped by echolocation calls and social calls" has, in its current form, been rejected for publication in Proceedings B.

This action has been taken on the advice of referees, who have recommended that substantial revisions are necessary. With this in mind we would be happy to consider a resubmission,

provided the comments of the referees are fully addressed. However please note that this is not a provisional acceptance.

Sincerely,
Dr Sasha Dall
<mailto:proceedingsb@royalsociety.org>

Associate Editor
Board Member: 1
Comments to Author:

Thank you for submitting your manuscript to Proceedings B. We have now received two reviews. Both of the reviewers found the topic and data interesting and valuable. However, one of the reviewers expressed strong concerns about the novelty of the main findings and also whether the conclusions of the two more unique results of the study are adequately supported by the data (sex differences) or the methods used to test the hypothesis (dynamic range differences).

Specifically, the reviewer requests an explanation of how the results in this study build on the results of Bohn et al. 2006, which also shows phylogenetically-controlled correlations between best hearing frequency and echolocation or social call frequency. The reviewer is also concerned that a sample size of three bats per sex is not sufficient to assess hearing differences between the sexes, especially considering the variation in the data.

Perhaps more importantly, the reviewer questions the assumption that changes in the amplitude of ABR recordings accurately reflects the dynamic range of hearing. Given the nature of the recording (compound response of many neurons firing), slight differences in neural synchrony and recruitment could have significant impacts on the summed amplitude. Is there a published study that can confirm that ABS amplitude is reliably correlated with neural activity such that it can be used to assess dynamic range and that this is consistent across frequencies? This was one of the most interesting results of the study and needs to be properly supported.

Both reviewers also provide valuable feedback that will improve a future version of this manuscript.

Reviewer(s)' Comments to Author:

Referee: 1

Comments to the Author(s)

The manuscript uses a combination of new ABR recordings from 11 species of bats with a meta-analyses of published data to explore two different but related questions: The first asks whether or not the auditory sensitivity range of a bat is influenced by both echolocation and social calls, and secondly whether the dynamic range of coding amplitude is different for high (echolocation) versus low (social call) frequency sounds. The introduction also raised a question about whether low-frequency hearing is influenced by the need to detect social calls or rather prey-generated noises (line 84). As it relates to the first question, the paper seems to come to very nearly the same conclusion of a similar paper (citation 11, Bohn et al., 2006) that also used meta-analyses to estimate the relative influence of social calls and echolocation on evolution of hearing thresholds. It is not made very clear what new conclusions are drawn from this newer analysis. The second question, that of dynamic range coding, has not been explored comparatively, and is a new and noteworthy idea. However, the paper makes it very difficult to discern what is actually meant by this, and it took me several reads through to understand how this was related to echolocation and communication. I think the average reader will struggle getting through the abstract.

While ABRs are widely used as a comparative tool for studying hearing thresholds between species, I am unaware of this tool being used to evaluate dynamic range, and after reading the paper I am not convinced that ABRs are a reliable way to do this. ABRs are complex summated potentials that may not be linearly or directly related to signal amplitude, because they reflect both an increase in individual neuron activity and also the total number of neurons excited. This is problematic if the auditory system has an acoustic fovea composed of a disproportionately larger number of neurons processing a particular important frequency range, and may not be directly related to minimum thresholds or amplitude coding per se at every frequency. So, at the very least the manuscript needs to build support for using ABRs to answer this question. Also, the manuscript mentions identifying “overarching principles or amplitude coding”, but doesn’t provide any background into what this means for mammals, vertebrates, or auditory systems overall. More generally, the rationale for the hypothesis that social calls might require less dynamic range than echolocation calls is not convincing.

Specific comments:

Line 160-162: Is this a new way of calculating dynamic range, or is this an established method in the field? Has it been done using ABR data before?

Line 242 and Figure 1. “The two most prominent peaks in the first isoresponse line...” Why do the peaks only show up in the first line, but not in the mean ABR threshold line or the second line? Similarly, why isn’t there a sensitivity peak at 60 kHz for the *Pteronotus parnellii* graph? Isn’t this species also well known for using multiple harmonics, at 30 and 90, and 120 kHz? For *P. parnellii* the first isoresponse line indicates rather poor sensitivity in the region where they should have the greatest sensitivity, which may reflect the limitations of using ABRs in this way. A quick review of literature shows that some of these ABR graphs are not very consistent with the literature (see for example Grinnell 1970, JCP 68:117-153). The text should address these discrepancies.

Figure 2. Superficially, this should be a simple transformation, and yet the data in figure 2 raise several questions. First, if the ABR threshold is very high, then this would automatically constrain the growth function, no? Line 161-162 says that a shallow slope represents a large dynamic range, and vice versa. Choosing *Molossus molossus* as an example, it is confusing why in fig 2 this animal’s dynamic range is highest above 50 kHz, and yet fig 1 shows that it can barely even detect frequencies about 50 kHz. The slope for *P. parnellii* similarly raises questions, because

it seems unlikely that for this animal the highest dynamic range would not be around its auditory fovea at 60 kHz.

Figure 3: The authors correctly note that ABR thresholds are not as low as those obtained with behavioral audiograms. ABRs vary considerably between labs, experimental conditions, and hardware being used. I will simply state that an $n=3$ per sex is probably insufficient to determine a sex difference in ABR thresholds. This point is only loosely related to the rest of the manuscript is not essential to the question of amplitude coding. Behavioral audiograms showed that male vampire bats displayed auditory thresholds 40-60 dB less than those shown here (Heffner et al 2013, *Hearing Res.*, 296: 42-50), so it is hard to believe that male vampire bats are truly 20-30 dB less sensitive than females, especially across their entire hearing range. Recommend removing this section from manuscript and focusing on amplitude coding.

Referee: 2

Comments to the Author(s)

One goal of this interesting manuscript (ms) was to measure the evoked auditory brainstem response (ABR) in 11 Neotropical bat species in response to both acoustic clicks and pure tones. Tonal ABRs were used to construct threshold tuning curves (audiograms), and suprathreshold ABR waveforms were used to examine amplitude coding within a frequency (i.e. the growth of the ABR waveform at different frequency-amplitude combinations), with a special focus comparing the low and high frequency hearing ranges across species: these ranges correspond to pup isolation calls and adult echolocation calls, respectively. The ms also looked at sex differences in ABR thresholds in four bat species where at least three individuals were measured, and found evidence that females had lower ABR thresholds at lower frequencies (where pups emit isolation calls) in two species but not in two others. The ms conducted a further comparative analysis by combining their ABR data with audiogram/hearing information obtained from the literature for an additional 27 bat species where the animals were tested with a variety of auditory electrophysiological recording techniques. The comparative analysis found that species-specific peaks in auditory sensitivity correlated reasonably well with the peak spectral frequencies of echolocation calls and pup isolation calls for that species. The ms concludes by suggesting that changes in hearing sensitivity evolved with changes in the frequency content of echolocation and social calls, and highlights the importance of social communication as an evolutionary pressure acting on auditory perception in bats.

Overall, I enjoyed the science in this ms. The results were noteworthy and important. I agree that this is one of the most comprehensive comparative assessments of the hearing capabilities of bats. Different species appear to have some differences in the overall shape of their mean (and 95% confidence intervals) clicked-evoked ABR waveforms presented at a very loud level (100 dB SPL); however, a between-species comparison was not conducted on these data. The overall shape of the tone-evoked ABR audiograms were somewhat similar but also showed some differences across species, but again there was no quantitative comparison of absolute thresholds or audiogram shape within or between species. There were some differences in the shape and strength of the averaged ABR signals at each frequency-amplitude combination, observed as different distances between bootstrapped iso-response contour lines within an audiogram and plotted as ABR growth functions. The ms reports that ABR growth functions at high frequencies (corresponding to the spectral range of echolocation calls) had shallower slopes compared to the generally steeper sloped ABR growth functions measured at lower frequencies (corresponding to the spectral range of pup isolation calls).

I appreciate the great effort that went into collecting and analyzing these data, which in general are of very high quality. The authors should be congratulated. One thing I did not enjoy was the large amount of information sequestered into the Electronic Supplementary Material. In general, I am never a fan of papers where there is more information in the Supplementary Material than in the paper itself. This seems to defeat the purpose of a stand-alone article. If the journal will allow, I encourage the authors to move all of the relevant and important details of ABR recording, signal

processing, and data analysis into the ms proper because the majority of the ms relies on these data. There were a few places where the text was unclear or confusing and I have highlighted them (see Specific Comments) with a goal to improve the flow and overall readability of the ms.

Lastly, all of the figures and tables are of high quality. In my opinion, Figures 1 and 2 are too small and/or over complicated and this takes away from the important information presented in them.

SPECIFIC COMMENTS

Introduction

1. (lines 38-46) It is difficult to separate the relative importance of the role of perceptual challenges associated with different echolocation niches and species phylogenetic origin. Other important factors related to the evolution (phylogeny) and perception (physics and neurobiology) of the spectral design of echolocation and social calls are beam shape of the signal and animal body size (e.g. Thiagavel et al. 2017, 2018; Barclay and Brigham 1991).

2. (lines 63-71) I agree that social calls are often emitted in close-proximity to conspecifics and heterospecifics. But comparing the functional ranges of these two general classes of vocalizations is difficult. If social signals are generally lower in frequency than echolocation calls, then they will attenuate more slowly, propagate further, and this could extend their functional range in comparison to that of echolocation signals (5-20 m). For example, some lekking bats emit vocalizations to attract mates, likely from distances further away than the functional range of echolocation. And except for a few species (e.g. *Desmodus rotundus*, *Diaemus youngi*, and *Diphylla ecaudata*), it remains unknown if bats emit social calls to maintain contact with nearby conspecifics and/or to guide their offspring during migrations or to new roost locations (e.g. Ripperger et al. *Biology Letters* 15(2): 20180884).

3. (lines 68-71) Minor point. This is an interesting idea, but I encourage the authors to consider editing this sentence so it does not presuppose that “bats do not need to perceive the echoes of their own social vocalizations.” I am unaware of evidence that bats use/do not use echoes of social vocalizations to obtain information (but I also know that bats never cease to amaze bat researchers).

4. (lines 83-85) A non-mutually exclusive alternative possibility to why bats have low-frequency hearing (in addition to listening for prey-generated sounds or pup isolation calls) is that sensitivity to low frequencies was crucial and selected for in the earliest mammals for general alertness and detecting predator-generated sounds (e.g. see Fig. 1 in Fullard 1988 *Experientia* 44, 423-428). Perhaps this is what was meant by saying this “may be a remnant from the bat’s phylogenetic ancestors” but the word remnant is tricky because it can suggest that an ability is a relic rather than still functionally useful and adaptive.

Methods

5. (lines 101-109) All of the data were collected in Panama but the ms does not say where... indoors in a lab/building or in the field (i.e. more details please)? The ms proper has more information and space devoted to the phylogenetic analysis than the auditory analysis. Because most of the Results rely on the ABR data, I think the methodological information about ABR electrophysiology and recording procedures should be moved into the main body of the ms because it is too important to be relegated to Supplementary Material.

6. (lines 112-115) The ms presents clicked-evoked ABRs measured at a high sound pressure level but does not say why click ABRs were collected and their value to the results and data analysis? Figure S3 shows that the mean click-evoked ABR waveforms at 100 dB SPL were quite variable within and between bat species, but there was no further analysis demonstrating whether this variation was meaningful/indicative or predictive of the species. Moreover, there are many sources of variation that could influence the strength and appearance of an ABR waveform (e.g.

differences in anesthesia state, electrode placement, skull thickness, electrical noise at time of recording, etc.). Clicked-evoked thresholds were ~60 dB peSPL. Is this because the clicks were shorter in duration and had less energy than the 2.5 ms pure tones? I encourage the authors to add a bit more clarifying text describing the value of collecting and presenting the click-evoked data in the ms given almost all of the ms focuses on the tone-evoked data.

7. (lines 154-157) This analysis assesses the growth in the suprathreshold ABR response with increasing stimulus level but does not compare ABR thresholds within or between a species (except later for a sex difference comparison in four species).

8. (lines 157-162) I was a bit confused by the description of the slope calculation. I assume that the plotted functions were the amplitudes of the evoked ABR signals at each SPL, and that the slope has units of $\mu\text{V}/\text{dB}$ (the units are missing in Fig. 2). Later (lines 160-161), the text suggests the dynamic range at a given frequency is the ratio of the loudest and faintest stimuli that can be detected. It would be helpful and less confusing to the reader if the definitions and calculations of the slope and the dynamic range of hearing were explicit.

Results

9. (lines 230-240) The Abstract says “ABR thresholds differed between species but showed a general tub-shaped trend” ... and later (lines 25-26) ... “In most species, auditory sensitivity was equally good at both high and low frequency ranges.” I mention this text from the Abstract here because the Results section did not compare ABR thresholds between the low and high frequency ranges within and/or between species. There was a comparison of ABR growth functions between the low (pup isolation calls) and high (echolocation calls) frequency ranges, and between males and females in 4 species, but not a comparison of thresholds. I’m not suggesting that the authors conduct a threshold comparison, but they should clarify the text of the Abstract and Results to make it clear that a comparative analysis of threshold tuning was not conducted.

10. (lines 235-236) Minor point. To help your readers, please specify the audiograms in Fig. 1 that have three (3) sensitivity peaks and whether they exist in the threshold tuning curves or in the bootstrapped iso-response lines.

11. (lines 252-254) The ms here could be clarified so that readers do not think the distance between iso-response lines is the ABR growth function. Also, please see point #8 on clarifying the calculation and slope units. I was confused by the word “flexible” to describe the two classes of ABR growth functions. How are ABR growth functions with shallower slopes more “flexible” than ABR growth functions with steeper slopes? My understanding is that growth functions with shallower slopes can encode changes in stimulus SPL more finely compared to growth functions with steeper slopes.

Discussion

12. (lines 300-304) I’m not sure I understand the point that is being made. Behavioral testing often uses a successive approximation procedure to measure thresholds, but I don’t understand why measuring ABRs is better for consistently testing responses to the same stimuli compared to using behavior to consistently test responses to the same stimuli? For example, both techniques can be consistently applied, both have multiple sources of variation, both can adapt/habituate, and both can fail. Perhaps I have misunderstood what is trying to be communicated?

13. (lines 351-352) Minor point. The results of Figure 4b were based on one type of social vocalization—pup isolation calls—so I suggest a slightly more conservative rewording of the second clause of this sentence so that it focuses on pup isolation calls that facilitate care and mother-offspring communication.

14. (lines 367-369) See point #11 about “flexibly” encoding SPL. What is meant by “auditory sensitivity”? Is this just absolute threshold? Figure 1 suggests that different species may not be equally sensitive in the low and high frequency ranges.

15. (Figure 1) This figure is critical, but the size of each plot is too small, and the panels seem overly complex. The numbers on the iso-response contour line are very small for my eyes, as are the red numbers above the x-axes. There is a wealth of color on this plot, including for other types of social vocalizations that were not the focus of the ms (i.e. is it necessary to show the bandwidths of other types of social calls when the ms is about echolocation versus pup isolation calls? Also, it is hard to tell the difference between red and pink at the small size)? I encourage the authors to make Figure 1 a full-page width, with taller and wider panels and larger font sizes.

16. (Figure 2) This figure plots the slope of the ABR growth function at each frequency. The units on the y-axis are not reported (i.e. $\mu\text{Vs}/\text{dB}$?). Again, each panel is too small. The R2 numbers are nearly impossible to read without magnification. The panels should be taller and wider, so the figure is a full-page width in the journal. The authors might consider standardizing the location of the bars representing the bandwidths of pup isolation calls and echolocation calls across all figures if they thought this would help improve readability.

17. (Figure 4) This is another very important figure. The caption for the legend says "Different symbols depict ways in which hearing thresholds were determined. SC: subcutaneous {potential} (ABRs), IC: inferior colliculus {potential} (electrophysiological recordings), PC = posterior colliculus {potential} (electrophysiological recordings)." The legend is a bit confusing because it is a mixture of electrophysiological recording techniques and anatomical locations. All three symbols (SC, IC, PC) refer to recording electrophysiological potentials but IC and PC reference to central auditory nuclei. The location of PC is undefined (i.e. does PC refer to the auditory thalamus and/or cortex or the auditory midbrain and/or brainstem?). Were all of the data collected with extracellular recording or were some from intracellular recording? The caption does not mention filled triangles and behavioral thresholds.

18. (Figure S2) Please specify if any of the data recorded with the higher frequency resolution technique were used in the final analysis (e.g. to measure ABR thresholds or to obtain ABR growth functions).

Author's Response to Decision Letter for (RSPB-2020-1170.R0)

See Appendix A.

RSPB-2020-2600.R0

Review form: Reviewer 1

Recommendation

Accept as is

Scientific importance: Is the manuscript an original and important contribution to its field?

Good

General interest: Is the paper of sufficient general interest?

Good

Quality of the paper: Is the overall quality of the paper suitable?

Excellent

Is the length of the paper justified?

Yes

Should the paper be seen by a specialist statistical reviewer?

No

Do you have any concerns about statistical analyses in this paper? If so, please specify them explicitly in your report.

No

It is a condition of publication that authors make their supporting data, code and materials available - either as supplementary material or hosted in an external repository. Please rate, if applicable, the supporting data on the following criteria.

Is it accessible?

Yes

Is it clear?

Yes

Is it adequate?

Yes

Do you have any ethical concerns with this paper?

No

Comments to the Author

The author's have addressed all my earlier concerns. The rationale supporting the methods is now much clearer, and the new version is significantly easier to understand. Thank you and congratulations on a very nice study.

Decision letter (RSPB-2020-2600.R0)

20-Nov-2020

Dear Dr Knörnschild

I am pleased to inform you that your Review manuscript RSPB-2020-2600 entitled "Hearing sensitivity and amplitude coding in bats are differentially shaped by echolocation calls and social calls" has been accepted for publication in Proceedings B.

The referee(s) do not recommend any further changes. Therefore, please proof-read your manuscript carefully and upload your final files for publication. Because the schedule for publication is very tight, it is a condition of publication that you submit the revised version of your manuscript within 7 days. If you do not think you will be able to meet this date please let me know immediately.

To upload your manuscript, log into <http://mc.manuscriptcentral.com/prsb> and enter your Author Centre, where you will find your manuscript title listed under "Manuscripts with Decisions." Under "Actions," click on "Create a Revision." Your manuscript number has been appended to denote a revision.

You will be unable to make your revisions on the originally submitted version of the manuscript. Instead, upload a new version through your Author Centre.

1) A text file of the manuscript (doc, txt, rtf or tex), including the references, tables (including captions) and figure captions. Please remove any tracked changes from the text before submission. PDF files are not an accepted format for the "Main Document".

2) A separate electronic file of each figure (tiff, EPS or print-quality PDF preferred). The format should be produced directly from original creation package, or original software format. Please note that PowerPoint files are not accepted.

3) Electronic supplementary material: this should be contained in a separate file from the main text and the file name should contain the author's name and journal name, e.g. `authorname_procb_ESM_figures.pdf`

All supplementary materials accompanying an accepted article will be treated as in their final form. They will be published alongside the paper on the journal website and posted on the online figshare repository. Files on figshare will be made available approximately one week before the accompanying article so that the supplementary material can be attributed a unique DOI. Please see: <https://royalsociety.org/journals/authors/author-guidelines/>

4) Data-Sharing and data citation

It is a condition of publication that data supporting your paper are made available. Data should be made available either in the electronic supplementary material or through an appropriate repository. Details of how to access data should be included in your paper. Please see <https://royalsociety.org/journals/ethics-policies/data-sharing-mining/> for more details.

<http://datadryad.org/submit?journalID=RSPB&manu=RSPB-2020-2600> which will take you to your unique entry in the Dryad repository.

Once again, thank you for submitting your manuscript to Proceedings B and I look forward to receiving your final version. If you have any questions at all, please do not hesitate to get in touch.

Sincerely,

Dr Sasha Dall

Associate Editor

Comments to Author:

Thank you for your careful and thorough response to the reviewer comments and the revision of your manuscript. Congratulations on a very interesting paper!

Reviewer(s)' Comments to Author:

Referee: 1

Comments to the Author(s).

The author's have addressed all my earlier concerns. The rationale supporting the methods is now much clearer, and the new version is significantly easier to understand. Thank you and congratulations on a very nice study.

Sincerely,
Proceedings B
mailto: proceedingsb@royalsociety.org

Associate Editor,
Comments to Author:

Thank you for your careful and thorough response to the reviewer comments and the revision of your manuscript. Congratulations on a very interesting paper!

Reviewer(s)' Comments to Author:

Referee: 1

Comments to the Author(s).

The author's have addressed all my earlier concerns. The rationale supporting the methods is now much clearer, and the new version is significantly easier to understand. Thank you and congratulations on a very nice study.

Decision letter (RSPB-2020-2600.R1)

30-Nov-2020

Dear Dr Knörnschild

I am pleased to inform you that your manuscript entitled "Hearing sensitivity and amplitude coding in bats are differentially shaped by echolocation calls and social calls" has been accepted for publication in Proceedings B.

Open Access

You are invited to opt for Open Access, making your freely available to all as soon as it is ready for publication under a CC BY licence. Our article processing charge for Open Access is £1700. Corresponding authors from member institutions (<http://royalsocietypublishing.org/site/librarians/allmembers.xhtml>) receive a 25% discount to these charges. For more information please visit <http://royalsocietypublishing.org/open-access>.

Paper charges

Sincerely,

Appendix A

Responses to the two reviews of the manuscript RSPB-2020-1170 entitled "Hearing sensitivity and amplitude coding in bats are differentially shaped by echolocation calls and social calls" submitted to Proceedings B.

Dear Dr. Dall,

we are writing regarding our manuscript (MS ID#: RSPB-2020-1170) entitled "Hearing sensitivity and amplitude coding in bats are differentially shaped by echolocation calls and social calls". We have now revised the manuscript and have addressed the comments of both reviewers in full. Their feedback was greatly appreciated, and the raised points have helped substantially to improve our manuscript in content and clarity. In light of their constructive feedback, we have incorporated all suggestions of the reviewers and rewritten the manuscript in parts.

In particular, we have (1) clarified the concept of the "dynamic range", its calculation and function, (2) reduced the focus on sexual dimorphic hearing in bats, (3) removed the redundant click-ABR data, and (4) instead included an additional figure summarizing the audiograms of all measured species in a direct comparative summary graphic. Moreover, we have (5) detailed how our work builds on the results of Bohn et al. 2006.

We believe that the revisions have improved the quality of our manuscript, and hope that you find that it now merits publication in the Proceedings of the Royal Society B. Thank you for handling the manuscript. We are looking forward to hearing from you.

Sincerely,
Mirjam Knörnschild

Comments to Authors

Associate Editor

COMMENT: Thank you for submitting your manuscript to Proceedings B. We have now received two reviews. Both of the reviewers found the topic and data interesting and valuable. However, one of the reviewers expressed strong concerns about the novelty of the main findings and also whether the conclusions of the two more unique results of the study are adequately supported by the data (sex differences) or the methods used to test the hypothesis (dynamic range differences).

Specifically, the reviewer requests an explanation of how the results in this study build on the results of Bohn et al. 2006, which also shows phylogenetically-controlled correlations between best hearing frequency and echolocation or social call frequency.

RESPONSE: Our study builds on the results of Bohn et al. 2006 in three important ways:

- 1) In contrast to Bohn et al. 2006, we use a modern phylogenetic comparative analysis that can account for the adaptive evolution of traits instead of simply modelling drift. We used a stochastic linear Ornstein-Uhlenbeck model of trait evolution (Hansen et al. 2008) that takes not only the Brownian motion-part of trait dynamics into account but also the rate of adaptation and the evolution toward an optimal state (as a correction for maladaptation). The latter two aspects make our model much better suited than phylogenetically independent contrasts - the model used by Bohn et al. 2006 - to investigate whether there was an adaptive correlated evolution between two traits, i.e. hearing sensitivity and call frequencies. Phylogenetically independent contrasts (originally proposed by Felsenstein in 1985) only rely on Brownian motion to model trait evolution and ignore the rate of adaptation and the optimum. Thus, while Brownian motion models are well suited for modelling drift, they cannot account for the adaptive evolution of traits (see Huey et al. 2019 for a detailed statement). In the revision, we emphasize the advantages of Ornstein-Uhlenbeck models to study the adaptive evolution of traits.
- 2) We used almost 3 times more data in our phylogenetic comparative analysis than Bohn et al. 2006, namely 38 species from 13 families (compared to 13 species from 6 families in Bohn et al. 2006), thus providing unprecedented evidence for the correlated evolution between hearing sensitivity and call frequencies in bats. Moreover, we contributed original data on hearing thresholds and call frequencies for 11 of the 38 species in our analysis (remaining data was taken from the literature); for 7 species, no information on hearing thresholds had

Responses to the two reviews of the manuscript RSPB-2020-1170 entitled "Hearing sensitivity and amplitude coding in bats are differentially shaped by echolocation calls and social calls" submitted to Proceedings B.

been available before. Our study is one of the most comprehensive comparative assessments of the hearing capabilities of bats available to date and we are confident that it will be the foundation for many future studies on acoustic production and perception in bats (see also the statement of referee 2 who fully agrees with us on this point).

- 3) We not only assessed hearing sensitivity but also stimulus level coding in high and low frequency ranges. Our results show that, for most species, auditory sensitivity was equally good at both high and low frequency ranges, while amplitude was more finely coded for higher frequencies (of echolocation calls) than for lower frequencies (of social calls). This difference in amplitude coding is probably related to the bats' need to finely encode the wide range of amplitude differences in echolocation calls and their echoes.

In the revision, we explain to what extent our study builds on the results of Bohn et al. (2006). For details, please see the answers to the referees' comments below.

COMMENT: The reviewer is also concerned that a sample size of three bats per sex is not sufficient to assess hearing differences between the sexes, especially considering the variation in the data.

RESPONSE: We completely agree with referee 1 that our sample size of 3-7 bats per sex and species is not large enough to assess sex-specific differences in hearing with certainty. Nevertheless, we want to mention this preliminary but potentially interesting result since we think it warrants further studies. Sex-specific differences in perception, irrespective of modality, are rarely studied in bats and we hope to provide an incentive for doing so in the future. In the revision, we rephrased the relevant sections to emphasize that our sex-specific results are preliminary and based on small sample sizes. Moreover, we moved the respective figure to the supporting information and shortened the discussion so that this result is not as prominent anymore as it was before.

COMMENT: Perhaps more importantly, the reviewer questions the assumption that changes in the amplitude of ABR recordings accurately reflects the dynamic range of hearing. Given the nature of the recording (compound response of many neurons firing), slight differences in neural synchrony and recruitment could have significant impacts on the summed amplitude. Is there a published study that can confirm that ABR amplitude is reliably correlated with neural activity such that it can be used to assess dynamic range and that this is consistent across frequencies? This was one of the most interesting results of the study and needs to be properly supported.

RESPONSE: The idea of using ABRs and their growth functions to evaluate dynamic range is indeed not a widely applied method but there are several published studies using this method. In the revised version of the manuscript, we have clarified the usefulness of ABRs and their growth functions for the dynamic range assessment in different species and provided more examples for previous studies using this method.

Generally, ABRs are widely used for the assessment of hearing thresholds (see for example Zheng et al., 1999; Liberman et al., 2014; Fischl et al., 2019). The dynamic range of hearing is only rarely investigated with the help of ABRs; however, the idea is not new: Grinnell (1970) measured ABR growth functions and correlated them with the bats' dynamic range of hearing.

The dynamic range is a common method to evaluate loudness perception in humans (see for example Serpanos et al., 1997). While the dynamic range assessed behaviourally in humans is immense (~120 dB), the dynamic range demonstrated with the use of ABRs is smaller (about 30-60 dB) (Simpson & Reiss, 2013) but, nevertheless, this shows that ABRs are indeed a useful measure for the assessment of the dynamic range of hearing. Aside from human psychophysical experiments, dynamic ranges are assessed in selected model animals. For example, in case of peripheral hearing deficits (due to cochlear impairment), ABR growth functions are used to assess the effect of the disruption on hearing (Möhrle et al., 2016).

The editor and referee 1 pointed out the nature of ABR recordings, which are compound response of many neurons firing, and have expressed concerns about the accuracy of reflection of the dynamic range of hearing. "ABRs are complex summated potentials that may not be linearly or directly related to signal amplitude, because they reflect both an increase in individual neuron

Responses to the two reviews of the manuscript RSPB-2020-1170 entitled "Hearing sensitivity and amplitude coding in bats are differentially shaped by echolocation calls and social calls" submitted to Proceedings B.

activity and also the total number of neurons excited" (as correctly stated by referee 1). This is the very point we want to highlight in our response and in the manuscript. The ABR measurement we used is a summed potential elicited with a tone-pip with a specific amplitude of at a certain frequency. Different amplitudes of this summed potential, apart from frequency and level, can indeed have various underlying causes: e.g. the number of neurons, the neuronal firing activity, or the nature of the afferent fibres (assuming stable physiological conditions during the experiment). ABRs are a tool to indicate these differences between frequencies, but they cannot be used to dissect the causes of these differences. The identification of these differences in signal strength at different frequencies can then be used for the extraction of an audiogram and the ABR growth function. In the manuscript, we hypothesise that the occurrence of shallow growth functions, presumably related to a wider dynamic range, at high frequency ranges is related to the bats' need to be able to finely encode the wide range of level differences of their echolocation calls. The ABRs thus allow us to extract a shallow ABR growth function slope, but they cannot be used to indicate the anatomical basis for this difference when compared to growth functions at lower frequency ranges. We have clarified the use and the limitations of the ABR measurements for hearing assessments in the revised version of the manuscript.

COMMENT: Both reviewers also provide valuable feedback that will improve a future version of this manuscript.

RESPONSE: The referees' feedback was greatly appreciated and helped us to improve our manuscript considerably. We have included detailed responses to the comments of both reviewers to demonstrate how we have addressed each of the points raised. Line numbers refer to the revised manuscript. Changes in the revised manuscript are highlighted.

Referee: 1

COMMENT: The manuscript uses a combination of new ABR recordings from 11 species of bats with a meta-analysis of published data to explore two different but related questions: The first asks whether or not the auditory sensitivity range of a bat is influenced by both echolocation and social calls, and secondly whether the dynamic range of coding amplitude is different for high (echolocation) versus low (social call) frequency sounds. The introduction also raised a question about whether low-frequency hearing is influenced by the need to detect social calls or rather prey-generated noises (line 84). As it relates to the first question, the paper seems to come to very nearly the same conclusion of a similar paper (citation 11, Bohn et al., 2006) that also used meta-analysis to estimate the relative influence of social calls and echolocation on evolution of hearing thresholds. It is not made very clear what new conclusions are drawn from this newer analysis.

RESPONSE: We are glad to know that the referee finds our study interesting. We apologize for not having stated clearer how our results build on the study of Bohn et al. 2006. In the revision, we emphasize

- 1) that we used a statistical model of trait evolution that is better suited for the study of correlated evolutionary processes since it can account for the adaptive evolution of traits,
- 2) that we increased the number of species in the analysis by almost 3 times, thus providing unprecedented evidence for the adaptive correlated evolution between hearing sensitivity and call frequencies in bats,
- 3) and that we not only assessed hearing sensitivity but also stimulus level coding in high and low frequency ranges.

Details to 1) We conducted a modern phylogenetic comparative analysis that can account for the adaptive evolution of traits instead of simply modelling drift. Our stochastic linear Ornstein-Uhlenbeck model of trait evolution (Hansen et al. 2008) is much better suited than phylogenetically independent contrasts - the model used by Bohn et al. 2006 - to investigate whether there was an adaptive correlated evolution between two traits, i.e. hearing sensitivity and call frequencies.

Phylogenetically independent contrasts (originally proposed by Felsenstein in 1985) only rely on Brownian motion to model trait evolution and ignore the rate of adaptation and the optimum.

Responses to the two reviews of the manuscript RSPB-2020-1170 entitled "Hearing sensitivity and amplitude coding in bats are differentially shaped by echolocation calls and social calls" submitted to Proceedings B.

Thus, while Brownian motion models are well suited for modelling drift, they cannot account for the adaptive evolution of traits (see Huey et al. 2019 for a detailed statement). Ornstein-Uhlenbeck models, on the other hand, take not only the Brownian motion-part of trait dynamics into account but also the rate of adaptation and the evolution toward an optimal state (as a correction for maladaptation).

We included information on the advantages of Ornstein-Uhlenbeck models of trait evolution in the revision.

Lines 196-201: Ornstein-Uhlenbeck models of trait evolution [29] not only model drift (Brownian motion-part of trait dynamics) but also the rate of adaptation and evolution of a trait (e.g. hearing sensitivity) toward an optimal state (as a linear function of a predictor; e.g. peak frequency of calls); this makes them well suited to test if there was an adaptive correlated evolution between two traits such as hearing sensitivity and call frequencies.

Details to 2) We used almost 3 times more data in our phylogenetic comparative analysis than Bohn et al. 2006, namely 38 species from 13 families (compared to 13 species from 6 families in Bohn et al. 2006), thus providing unprecedented evidence for the correlated evolution between hearing sensitivity and call frequencies in bats. Moreover, we contributed original data on hearing thresholds and call frequencies for 11 of the 38 species in our analysis (remaining data was taken from the literature); for 7 species, no information on hearing thresholds had been available before. In the revision, we state that we considerably expanded the data set used for comparative analyses.

Line 101: For seven species, no information on hearing thresholds had been available before.

Lines 183-186: In total, our analysis included 38 species from 13 families (echolocation call data: 37 species; isolation call data: 27 species). We thus conducted the phylogenetic comparative analysis with considerably more data than a previous study (Bohn et al. 2006), which included 13 species from 6 families.

Details to 3) In contrast to Bohn et al. (2006), we not only assessed hearing sensitivity but also stimulus level coding in high and low frequency ranges (by calculating the ABR growth function). Our results show that, for most species, auditory sensitivity was equally good at both high and low frequency ranges, while amplitude was more finely coded for higher frequencies (of echolocation calls) than for lower frequencies (of social vocalizations). This difference in amplitude coding is probably related to the bats' need to finely encode the wide range of amplitude differences in echolocation calls and their echoes.

Lines 314-317: In this study, we measured ABRs of 86 bats from 11 species and compiled data on 27 additional bat species from the literature. For 11 species, we not only assessed hearing sensitivity but also stimulus level coding in high and low frequency ranges (by calculating the ABR growth function). This provides the most comprehensive comparative assessment of the hearing capacity of bats to date.

Taken together, our manuscript provides a large body of evidence complementary to the findings of Bohn et al. 2006 and, more importantly, a so far lacking in-depth phylogenetic analysis illustrating that hearing sensitivity peaks in bats evolved in correlation with the species-specific frequency ranges of echolocation calls and social vocalizations. Moreover, our study shows that amplitude is more finely coded for higher frequencies (corresponding to echolocation calls) than for lower frequencies (corresponding for social vocalizations).

COMMENT: The second question, that of dynamic range coding, has not been explored comparatively, and is a new and noteworthy idea. However, the paper makes it very difficult to discern what is actually meant by this, and it took me several reads through to understand how this was related to echolocation and communication. I think the average reader will struggle getting through the abstract.

RESPONSE: We are grateful that the referee considers the novelty and findings of our dynamic range investigation noteworthy. We agree with the referee that the earlier version of our manuscript was not direct and clear enough on the definitions of the dynamic range concept and

how it relates to the bats' echolocation and communication. We have now clarified the concept of dynamic range and amplitude coding in the abstract, introduction, methods, and discussion. Most importantly, we are more careful in our correlation between the ABR growth functions and the dynamic range of hearing.

Lines 20-24: We tested if hearing sensitivity and stimulus level coding in bats differ between high and low frequency ranges by measuring auditory brainstem responses (ABRs) of 86 bats belonging to 11 species. In most species, auditory sensitivity was equally good at both high and low frequency ranges, while amplitude was more finely coded for higher frequencies.

Lines 68-73: Auditory brainstem responses (ABRs) are acoustically evoked summed electrical potentials that have been established as a fast, objective, and minimally invasive assessment of hearing in different species since the 1970s [24, 25]. Furthermore, the ABR growth functions can be related to the dynamic range of hearing, as has been demonstrated for example in bats [26], rats [27], and loudness perception in humans [28], and can thus be used to assess signal level encoding in the auditory pathway.

Lines 171-178: To calculate the slope of growth functions at each stimulus frequency, a sigmoidal curve was fitted to each function (using the `nlinfit` function in Matlab) and the coefficient of determination (R^2) of the fits, indicating the goodness, was assessed. A shallow slope suggests a large dynamic range, i.e. stimulus level differences are finely coded at this frequency, while a steep slope indicates a smaller dynamic range. Using ABR growth functions to assess the dynamic range of hearing, i.e. the ratio between the loudest and faintest stimuli that can be detected, has mainly been explored in relation to human loudness perception in the past [28, 37, 38], but has also been applied in animal models [26, 27].

COMMENT: While ABRs are widely used as a comparative tool for studying hearing thresholds between species, I am unaware of this tool being used to evaluate dynamic range, and after reading the paper I am not convinced that ABRs are a reliable way to do this. ABRs are complex summated potentials that may not be linearly or directly related to signal amplitude, because they reflect both an increase in individual neuron activity and also the total number of neurons excited. This is problematic if the auditory system has an acoustic fovea composed of a disproportionately larger number of neurons processing a particular important frequency range, and may not be directly related to minimum thresholds or amplitude coding per se at every frequency. So, at the very least the manuscript needs to build support for using ABRs to answer this question.

RESPONSE: The idea of using ABRs and their growth functions to evaluate dynamic range is indeed not a widely applied method, and we understand the referee's concerns. Generally, ABRs are widely used for the assessment of hearing thresholds (see for example Zheng et al., 1999; Liberman et al., 2014; Fischl et al., 2019). The dynamic range of hearing is only rarely investigated with the help of ABRs; however, the idea is not new: Grinnell (1970) measured ABR growth functions and correlated them with the bats' dynamic range of hearing.

The dynamic range is a common, objective method to evaluate loudness perception in humans, see for example (Serpanos et al., 1997). While the dynamic range assessed behaviourally in humans is immense (~120 dB), the dynamic range demonstrated with the use of ABRs is smaller (about 30-60 dB) (Simpson & Reiss, 2013) but, nevertheless, this shows that ABRs are indeed a useful measure for the assessment of the dynamic range of hearing. Aside from human psychophysical experiments, dynamic ranges are assessed in selected model animals. For example, in case of peripheral hearing deficits (due to cochlear impairment), ABR growth functions are used to assess the effect of the disruption on hearing (Möhrle et al., 2016). In the revised version of the manuscript, we have clarified the usefulness of ABRs and their growth functions for the dynamic range assessment in different species and provided more examples for previous studies using this method.

Lines 68-73: Auditory brainstem responses (ABRs) are acoustically evoked summed electrical potentials that have been established as a fast, objective, and minimally invasive assessment of hearing in different species since the 1970s [24, 25]. Furthermore, the ABR growth functions can be related to the dynamic range of hearing, as has been demonstrated for example in bats [26],

Responses to the two reviews of the manuscript RSPB-2020-1170 entitled "Hearing sensitivity and amplitude coding in bats are differentially shaped by echolocation calls and social calls" submitted to Proceedings B.

rats [27], and loudness perception in humans [28], and can thus be used to assess signal level encoding in the auditory pathway.

The referee raises an interesting point when they state that “ABRs are complex summated potentials that may not be linearly or directly related to signal amplitude, because they reflect both an increase in individual neuron activity and also the total number of neurons excited”. This is indeed the case and it is the very point we want to highlight in our response and in the manuscript. The ABR measurement we use is a summed potential elicited with a tone-pip with a specific level at a certain frequency. Different amplitudes of this summed potential, apart from frequency and level, can indeed have various underlying causes: e.g. the number of neurons, the neuronal firing activity, or the nature of the afferent fibres (assuming stable physiological conditions during the experiment). ABRs are a tool to indicate these differences between frequencies, but they cannot be used to dissect the causes of these differences. The identification of these differences in signal strength at different frequencies can then be used for the extraction of an audiogram and the ABR growth function.

In our manuscript, we hypothesise that the occurrence of shallow growth functions, presumably related to a wider dynamic range, at high frequency ranges is related to the bats’ need to be able to finely encode the wide range of level differences of their echolocation calls. The ABRs thus allow us to extract a shallow ABR growth function slope, but they cannot be used to indicate the anatomical basis for this difference when compared to growth functions at lower frequency ranges. We have clarified the use and the limitations of the ABR measurements for hearing assessments in the revised version of the manuscript.

Lines 86-90: We also assessed the dynamic range of hearing in high and low frequency ranges by calculating the magnitude of the supra-threshold ABRs for each species; this allowed us to identify shared principles of stimulus level coding between the measured species. We hypothesized that the coding of level differences would be more resolved for higher frequency ranges than for lower frequency ranges.

The referee also mentions the problematic nature of ABR measurements in the presence of an acoustic fovea. An acoustic fovea has been demonstrated in the cochleae of some bat species (specifically the species *P. parnellii* tested in our study). As stated above, the occurrence of different innervation patterns or a different processing of acoustic signals in the fovea is not a problem intrinsic to ABR measurements. However, as discussed in the detail in the supplementary material, in the present study we used the same stimulus matrix for all bat species, which likely caused us to sample ABR signals from regions around the fovea. This issue is not related to the concerns raised by the referee in this section and we respond to this issue in more detail in our response [three questions] below.

*Page 16, ESM: The audiogram of *P. parnellii* shows a sensitivity peak corresponding to the CF call bandwidth, which is very narrow and corresponds to their acoustic fovea (Kössl & Vater, 1985; Kössl 1994). In this study, we used a predefined parameter space (see methods), which we tested in all bat species. Although we detected a peak in the ABR threshold, which corresponds to the main CF component of *P. parnellii*, we most likely did not measure at the exact position of the individual CF bandwidth and thus missed the most sensitive point of hearing. Furthermore, the current tone-pips may be spectrally too broad to probe the narrow-band sensitivity changes in *P. parnellii* hearing around their main CF component.*

COMMENT: Also, the manuscript mentions identifying “overarching principles of amplitude coding”, but doesn’t provide any background into what this means for mammals, vertebrates, or auditory systems overall. More generally, the rationale for the hypothesis that social calls might require less dynamic range than echolocation calls is not convincing.

RESPONSE: In our manuscript, we show that there is a general trend in all tested bat species for shallower ABR growth functions in high frequency ranges (especially in the range corresponding to the echolocation call frequencies of each bat species) and steeper ABR growth functions in lower frequency ranges (especially those corresponding to the communication call range of the respective bat species). This trend is apparent in all 11 tested bat species, which led us to claim an

Responses to the two reviews of the manuscript RSPB-2020-1170 entitled "Hearing sensitivity and amplitude coding in bats are differentially shaped by echolocation calls and social calls" submitted to Proceedings B.

“overarching principle”. We have now changed the terminology to “shared” and “general principle”, which are less presumptuous, and have specified that our findings are specifically concerned with the measured bat species.

Lines 388-392: Our large-scale comparison of hearing capacities in bats not only allowed us to investigate commonalities and differences between species, but also to identify a species-independent, overarching principle for the perception of different signal types. Amplitude is more finely encoded in the high frequency range of echolocation calls than in the low frequency range of social calls, while auditory sensitivity is equally good at both high and low frequency ranges.

Lines 350-354: Our findings show that although the frequency range of echolocation calls and social vocalizations is species-specific, all eleven measured bat species show the same general principle of level coding: a shallow ABR growth function, i.e. large dynamic range of hearing, in the frequency range of their echolocation calls and a steep ABR growth function, i.e. small dynamic range of hearing, in the frequency range of their social vocalizations.

We agree with the referee that in the earlier version of our manuscript, we did not provide supporting information about the broader meaning of this principle for other species. We have now added a hypothesis for the generalization of this trend among laryngeally echolocating bat species and mentioned interesting future research avenues on that topic.

Lines 350-357: Our findings show that although the frequency range of echolocation calls and social vocalizations is species-specific, all eleven measured bat species show the same general principle of level coding: a shallow ABR growth function, i.e. large dynamic range of hearing, in the frequency range of their echolocation calls and a steep ABR growth function, i.e. small dynamic range of hearing, in the frequency range of their social vocalizations. It seems straightforward to assume that this is a shared principle between all laryngeally echolocating bats. It would be interesting to study this phenomenon in bat species that do not rely as strongly on the detection of minute level differences in a specific frequency range, such as Pteropodid bats.

SPECIFIC COMMENT: Line 160-162: Is this a new way of calculating dynamic range, or is this an established method in the field? Has it been done using ABR data before?

RESPONSE: ABRs have historically been used to assess hearing thresholds in anaesthetised animals, but only rarely have growth function been investigated. However, as mentioned earlier in this response, it is not an entirely new idea and the correlation between ABR growth functions and the dynamic range of hearing in bats has indeed been mentioned already 50 years ago (Grinnell, 1970). ABR dynamic ranges have mainly been explored in relation to human loudness perception (Serpanos et al., 1997; Silva & Epstein, 2010; Howe & Decker, 1984), but also in animal models (Möhrle et al., 2016). We have added this information in the manuscript as well.

Lines 175-178: Using ABR growth functions to assess the dynamic range of hearing, i.e. the ratio between the loudest and faintest stimuli that can be detected, has mainly been explored in relation to human loudness perception in the past [28, 37, 38], but has also been applied in animal models [26, 27].

SPECIFIC COMMENT: Line 242 and Figure 1. “The two most prominent peaks in the first isoresponse line...” Why do the peaks only show up in the first line, but not in the mean ABR threshold line or the second line? Similarly, why isn’t there a sensitivity peak at 60 kHz for the *Pteronotus parnellii* graph? Isn’t this species also well known for using multiple harmonics, at 30 and 90, and 120 kHz? For *P. parnellii* the first isoresponse line indicates rather poor sensitivity in the region where they should have the greatest sensitivity, which may reflect the limitations of using ABRs in this way.

RESPONSE: The isoresponse lines and the mean ABR threshold lines present two different types of information. The threshold is a calculated value indicating the lowest stimulus level where a statistically significant difference of the ABR signal from background noise occurred (containing no information about the ABR signal amplitude). The isoresponse lines indicate stimulus levels

across frequencies evoking the same ABR amplitude. This means that the threshold has no definite ABR amplitude value and can represent different ABR signal amplitudes for different frequencies, depending also on the noise floor. The isoresponse lines on the other hand indicate a specific measured value, which is independent from previous values (e.g. for different amplitudes at the same tested frequency). Therefore, the occurrence of peaks in the isoresponse lines are independent of peaks in other isoresponse lines and do not need to be in parallel to the mean ABR threshold. This is a very complex issue, which has been explained in more detail in the revised version of the manuscript.

Lines 164-170: The hearing threshold or audiogram is thus a calculated value indicating the lowest stimulus level where a statistically significant difference of the ABR signal from background noise occurred. The hearing threshold itself contains no information about the ABR signal amplitude and can represent different ABR signal amplitudes for different frequencies, depending also on the noise floor. The isoresponse lines are not bootstrapped, but instead indicate the species-specific average ABR signal strength (in μV) for each measured frequency-amplitude combination. Isoresponse lines are independent from previous values (e.g. for different amplitudes at the same tested frequency).

The referee is correct in their comment about the acoustic fovea of *P. parnellii* being at roughly 60 kHz and the expected peak in the ABR threshold. However, the exact location of the *P. parnellii* fovea is individually different, very narrow (~ 2 kHz), and can range from ~ 57 to ~ 64 kHz (Kössl & Vater, 1985). It is very likely that we missed measuring precisely at the fovea's characteristic frequency in most, if not all tested *P. parnellii* individuals, as test frequencies were not individualized. In this study we aimed to make our results comparable between species. Therefore, we tested the same set of frequencies for all bat species (11 logarithmically distributed frequency steps from 5 -120 kHz). This means, for example, that we measured ABRs evoked with a 63.5 kHz stimulus in all bat species and have furthermore clearly missed the fovea with the adjacent tested frequencies (i.e. 46.3 and 87.3 kHz). This study was designed for comparability between species, which comes at the cost that it does not reflect the particularities of the measured species. This is indeed one of the main limitations of this study and has been discussed in detail in the supplementary material.

Page 16, ESM: For Pteronotus parnellii, audiograms obtained with cochlear microphonics, i.e. evoked otoacoustic emissions, are available (Kössl & Vater 1985; Kössl 1994). Pteronotus parnellii is the only bat species measured in this study that produces constant frequency (CF) echolocation calls. CF calls have a very small bandwidth, but the echolocation frequencies are individually different and can vary by 35 Hz (Keating et al. 1994). The audiogram of P. parnellii shows a sensitivity peak corresponding to the CF call bandwidth, which is very narrow and corresponds to their acoustic fovea (Kössl & Vater 1985; Kössl 1994). In this study, we used a predefined parameter space (see methods), which we tested in all bat species. Although we detected a peak in the ABR threshold, which corresponds to the main CF component of P. parnellii, we most likely did not measure at the exact position of the individual CF bandwidth and thus missed the most sensitive point of hearing. Furthermore, the current tone-pips may be spectrally too broad to probe the narrow-band sensitivity changes in P. parnellii hearing around their main CF component.

COMMENT: A quick review of literature shows that some of these ABR graphs are not very consistent with the literature (see for example Grinnell 1970, JCP 68:117-153). The text should address these discrepancies.

RESPONSE: We agree with the referee that there are differences in the shape of the ABR thresholds in the study by Grinnell (1970) and the ones measured in the present study. We furthermore are aware that there are differences between the presented ABR thresholds and those measured in more recent studies (see "Comparison of new ABR data with existing audiograms in the literature" in the supplementary material). As referee 1 states further below: "ABRs vary considerably between labs, experimental conditions, and hardware being used." This is indeed the case, but it should be pointed out again that this is, in our opinion, one of the strengths of this paper. We compare many species with exactly the same methods - as opposed to comparing the

information from many papers dealing with only one or a few species, inherently acquired with different methods.

The specific differences between our data and the data presented in Grinnell's paper (1970) are based on two very important differences. Grinnell measured "just detectable differences" in the ABR amplitude (see methods section of the Grinnell (1970) paper), which at the time were likely assessed visually. The first difference is thus the applied criterion for ABR signal detection as we applied a statistical criterion (bootstrap analysis). It is likely that our automatically evaluated, statistical criterion led to higher thresholds than the ones documented by Grinnell, as it is more conservative. The second major difference lies in the analysed portion of the ABR signal: While Grinnell analysed the ABR strength on the basis of a single component (N4 = fourth ABR wave), we analysed the RMS of the complete signal, including all waves. We used the RMS as the waveforms of the ABR signals are quite variable between the species and we wanted to exclude the possibility of biasing towards individual species. Furthermore, the identification of specific ABR waves has been difficult in a number of studies (Beattie, 1988; Cebulla et al., 2014), which is a further argument supporting the use of the entire signal, rather than selected waves. The rationale behind using the RMS of the entire signal and the bootstrap criterion for the ABR threshold detection is now outlined in the manuscript.

Lines 147-152: The amplitudes of the recorded ABRs were calculated as the root-mean square (RMS) in the time window starting directly after the stimulus presentation and lasting for the duration of the ABR signal (i.e. 1–8 ms after stimulus onset). The RMS of the full signal was used in order to evade unreliability from individual waveform discrimination [33, 34] and to increase comparability with the recent literature [32, 35]. Bootstrap analyses (n = 500; 95% confidence) were performed on the ABR data to statistically verify the presence of an ABR signal [36].

We have discussed differences between the audiograms measured in the present study and ABRs measured in other recent studies in the supplementary material (see "Comparison of new ABR data with existing audiograms in the literature"). We have not included the ABRs measured by Grinnell (1970) in this comparison as they are, as outlined above, recorded with a partially outdated methodology.

SPECIFIC COMMENT: Figure 2. Superficially, this should be a simple transformation, and yet the data in figure 2 raise several questions. First, if the ABR threshold is very high, then this would automatically constrain the growth function, no?

RESPONSE: The referee is correct, that if a specific stimulus frequency results in a high threshold, fewer suprathreshold measuring points will be available for the assessment of the ABR growth function. However, this does not necessarily lead to a constriction in the values of the growth function itself, as the growth functions are based on the isoresponse contour lines. Rather, the reason for the low growth function slopes in high thresholds areas is that the data was acquired from the extremes of the tested frequency range (e.g. very high or low frequencies) and generally have a low R^2 for their slope fits as they are barely above the threshold, making them less reliable. We tried to make this transparent by providing the goodness of the fit, where it is low. However, high thresholds do not invariably lead to constrained growth function. Examples can be seen in Figure 2. *G. soricina* has a similar threshold at very high and low frequencies, however the isoresponse lines are differently dispersed (and thus the same is true for the ABR growth functions). A similar example can be seen for *D. rotundus*. We have clarified the difference and independence of the auditory threshold and the isoresponse lines in the revised manuscript.

Lines 164-170: The hearing threshold or audiogram is thus a calculated value indicating the lowest stimulus level where a statistically significant difference of the ABR signal from background noise occurred. The hearing threshold itself contains no information about the ABR signal amplitude and can represent different ABR signal amplitudes for different frequencies, depending also on the noise floor. The isoresponse lines are not bootstrapped, but instead indicate the species-specific average ABR signal strength (in μV) for each measured frequency-

Responses to the two reviews of the manuscript RSPB-2020-1170 entitled "Hearing sensitivity and amplitude coding in bats are differentially shaped by echolocation calls and social calls" submitted to Proceedings B.

amplitude combination. Isoresponse lines are independent from previous values (e.g. for different amplitudes at the same tested frequency).

SPECIFIC COMMENT: Line 161-162 says that a shallow slope represents a large dynamic range, and vice versa. Choosing *Molossus molossus* as an example, it is confusing why in fig 2 this animal's dynamic range is highest above 50 kHz, and yet fig 1 shows that it can barely even detect frequencies about 50 kHz. The slope for *P. parnellii* similarly raises questions, because it seems unlikely that for this animal the highest dynamic range would not be around its auditory fovea at 60 kHz.

RESPONSE: The audiogram for *M. molossus* is starting to show a sensitivity reduction around 50 kHz, but only reaches a region of strongly reduced hearing sensitivity at around 80 – 90 kHz (similar to other species tested here). This means *M. molossus* is certainly able to perceive acoustic signals well above 50 kHz (Fig. 1). The shallowest growth functions for this species were indeed calculated for stimulus frequencies of 87.3 and 120 kHz, however these are indicated with having a low R² value, and are therefore less reliable. The growth function in the echolocation call frequency of *P. parnellii* is, in accordance with our hypothesis, shallower than in the communication call frequency range (Fig. 2). It is interesting that the shallowest growth function, which is only slightly lower than the one in the echolocation range (Fig. 2), was measured at 24.5 kHz and not at 60 kHz, but this is most likely related to the fact that we do not use stimuli close to the animals' foveae, which in *P. parnellii* is very sharply tuned.

Page 16, EMS: In this study, we used a predefined parameter space (see methods), which we tested in all bat species. Although we detected a peak in the ABR threshold, which corresponds to the main CF component of P. parnellii, we most likely did not measure at the exact position of the individual CF bandwidth and thus missed the most sensitive point of hearing. Furthermore, the current tone-pips may be spectrally too broad to probe the narrow-band sensitivity changes in P. parnellii hearing around their main CF component.

SPECIFIC COMMENT: Figure 3: The authors correctly note that ABR thresholds are not as low as those obtained with behavioral audiograms. ABRs vary considerably between labs, experimental conditions, and hardware being used. I will simply state that an n=3 per sex is probably insufficient to determine a sex difference in ABR thresholds. This point is only loosely related to the rest of the manuscript is not essential to the question of amplitude coding. Behavioral audiograms showed that male vampire bats displayed auditory thresholds 40-60 dB less than those shown here (Heffner et al. 2013, Hearing Res., 296: 42-50), so it is hard to believe that male vampire bats are truly 20-30 dB less sensitive than females, especially across their entire hearing range. Recommend removing this section from manuscript and focusing on amplitude coding.

RESPONSE: We agree with the referee our sample size (3-7 individuals per sex and species) is not large enough to assess sex-specific differences in hearing with certainty. Nevertheless, we would prefer keeping the sex-specific analysis in the manuscript to indicate a potentially interesting future research avenue. However, we have amended the manuscript drastically in order to make it very clear that these results are only indicative of a trend and are not representing an absolute statement. We have deleted our hypotheses about sexual dimorphic hearing and our hypotheses about its function from the abstract and the introduction, moved the corresponding figure to the supplements, and reduced the extent to which we mention the topic in the discussion. We have further included the following additional disclaimer in the discussion.

Lines 367-370: These results need to be interpreted with care as the sample size for each sex is limited. Therefore, at present, we can only speculate whether sexual dimorphism in hearing sensitivity is commonly expressed in bats and which function it might serve.

However, we do think that this is an interesting finding and that future studies should investigate this promising research direction. Furthermore, this finding could be used as an argument for thorough analysis for potential sex bias in the data, which is an issue that recently gained a lot of attention in the scientific community (Beery & Zucker, 2011). This being said, we are prepared to remove this section from the manuscript in the future if the referee and editor still agree that it distracts too much from the main message.

Responses to the two reviews of the manuscript RSPB-2020-1170 entitled "Hearing sensitivity and amplitude coding in bats are differentially shaped by echolocation calls and social calls" submitted to Proceedings B.

Referee: 2

COMMENT: One goal of this interesting manuscript (ms) was to measure the evoked auditory brainstem response (ABR) in 11 Neotropical bat species in response to both acoustic clicks and pure tones. Tonal ABRs were used to construct threshold tuning curves (audiograms), and suprathreshold ABR waveforms were used to examine amplitude coding within a frequency (i.e. the growth of the ABR waveform at different frequency-amplitude combinations), with a special focus comparing the low and high frequency hearing ranges across species: these ranges correspond to pup isolation calls and adult echolocation calls, respectively. The ms also looked at sex differences in ABR thresholds in four bat species where at least three individuals were measured, and found evidence that females had lower ABR thresholds at lower frequencies (where pups emit isolation calls) in two species but not in two others. The ms conducted a further comparative analysis by combining their ABR data with audiogram/hearing information obtained from the literature for an additional 27 bat species where the animals were tested with a variety of auditory electrophysiological recording techniques. The comparative analysis found that species-specific peaks in auditory sensitivity correlated reasonably well with the peak spectral frequencies of echolocation calls and pup isolation calls for that species. The ms concludes by suggesting that changes in hearing sensitivity evolved with changes in the frequency content of echolocation and social calls, and highlights the importance of social communication as an evolutionary pressure acting on auditory perception in bats.

Overall, I enjoyed the science in this ms. The results were noteworthy and important. I agree that this is one of the most comprehensive comparative assessments of the hearing capabilities of bats.

RESPONSE: We are glad our manuscripts warrants such a positive perception and are grateful for the constructive criticism, which helped us to further improve this manuscript.

COMMENT: Different species appear to have some differences in the overall shape of their mean (and 95% confidence intervals) clicked-evoked ABR waveforms presented at a very loud level (100 dB SPL); however, a between-species comparison was not conducted on these data. The overall shape of the tone-evoked ABR audiograms were somewhat similar but also showed some differences across species, but again there was no quantitative comparison of absolute thresholds or audiogram shape within or between species. There were some differences in the shape and strength of the averaged ABR signals at each frequency-amplitude combination, observed as different distances between bootstrapped iso-response contour lines within an audiogram and plotted as ABR growth functions. The ms reports that ABR growth functions at high frequencies (corresponding to the spectral range of echolocation calls) had shallower slopes compared to the generally steeper sloped ABR growth functions measured at lower frequencies (corresponding to the spectral range of pup isolation calls).

RESPONSE: We take note from the referee that our within- and between-species comparison is not presented to a sufficient extent. Although the presentation of our data (especially the standard error of the mean (SEM)) is, in our opinion, a first step in the direction of a within species comparison, in the previous version of the manuscript we were not explicit enough in the discussion of these within species differences. The between species comparison has previously been conducted only in very limited comparisons in the discussion. In the revision, we have included an additional figure in the supplementary material, directly comparing all measured audiograms and extended the comparative section in the discussion. The detailed changes we made along the lines of a more elaborate comparative approach of bat hearing are listed below in answer to the specific comments of referee 2.

COMMENT: **I appreciate the great effort that went into collecting and analyzing these data, which in general are of very high quality. The authors should be congratulated.** One thing I did not enjoy was the large amount of information sequestered into the Electronic Supplementary Material. In general, I am never a fan of papers where there is more information in the Supplementary Material than in the paper itself. This seems to defeat the purpose of a stand-alone article. If the journal will allow, I encourage the authors to move all of the relevant and important details of ABR recording, signal processing, and data analysis into the ms proper because the majority of the ms relies

Responses to the two reviews of the manuscript RSPB-2020-1170 entitled "Hearing sensitivity and amplitude coding in bats are differentially shaped by echolocation calls and social calls" submitted to Proceedings B.

on these data. There were a few places where the text was unclear or confusing and I have highlighted them (see Specific Comments) with a goal to improve the flow and overall readability of the ms.

RESPONSE: We greatly appreciate the inclination of the referee to move more information from the supplementary material into the main text. We too would like to be more detailed in the main part of our ms. However, the journal requirements are very strict about word and figure number limits and we have to comply with these regulations. In order to still make use of this constructive comment, we have moved the ABR recording section from the supplementary material to the main text, thus better detailing the process of ABR acquisitions.

COMMENT: Lastly, all of the figures and tables are of high quality. In my opinion, Figures 1 and 2 are too small and/or over complicated and this takes away from the important information presented in them.

RESPONSE: Thank you for this helpful comment. We have rearranged the panels of figures 1 and 2 in order to give each panel more space. Both figures are now only two panels wide, which will allow the figures to take up more space when printed.

INTRODUCTION

COMMENT: 1. (lines 38-46) It is difficult to separate the relative importance of the role of perceptual challenges associated with different echolocation niches and species phylogenetic origin. Other important factors related to the evolution (phylogeny) and perception (physics and neurobiology) of the spectral design of echolocation and social calls are beam shape of the signal and animal body size (e.g. Thiagavel et al. 2017, 2018; Barclay and Brigham 1991).

RESPONSE: Thank you for pointing this out. We added information to emphasize that other factors also influence the spectral design of bat vocalizations.

Lines 40-42: However, separating the contributing effects of phylogeny and perception on call design is difficult since some factors, such as beam shape and body size, are shaped by both and can influence call design considerably as well [8].

COMMENT: 2. (lines 63-71) I agree that social calls are often emitted in close-proximity to conspecifics and heterospecifics. But comparing the functional ranges of these two general classes of vocalizations is difficult. If social signals are generally lower in frequency than echolocation calls, then they will attenuate more slowly, propagate further, and this could extend their functional range in comparison to that of echolocation signals (5-20 m). For example, some lekking bats emit vocalizations to attract mates, likely from distances further away than the functional range of echolocation. And except for a few species (e.g. *Desmodus rotundus*, *Diaemus youngi*, and *Diphylla ecaudata*), it remains unknown if bats emit social calls to maintain contact with nearby conspecifics and/or to guide their offspring during migrations or to new roost locations (e.g. Ripperger et al. *Biology Letters* 15(2): 20180884).

RESPONSE: This constructive comment has helped us to reword the introduction to be more precise about the functional differentiation of these call types. We phrased the differences in amplitude perception between social and echolocation calls in a more general way now and have included the referee's example in the introduction. We have now focused more on the information coded within the amplitude differences of these call types, instead of their perception.

Lines 56-66: Hearing in bats is adjusted in a species-specific way to the respective acoustic signal types and the situations in which they are produced [13]. However, hearing in bats should employ common principles to accommodate for the fundamental differences of echolocation calls and social vocalizations. Echolocation calls are generally produced at higher frequency ranges than social vocalizations in a given species [18]. Moreover, echolocation calls need to work over a broad range of distances to detect both near and far objects [3]. Thus, bats need to perceive both their own loud calls and their faint echoes [19] and sometimes even the echolocation calls of other bats [20]. On the other hand, social vocalizations are generally close-range signals that are typically perceived with similar intensities. Notable exceptions are lekking bat species, which

Responses to the two reviews of the manuscript RSPB-2020-1170 entitled "Hearing sensitivity and amplitude coding in bats are differentially shaped by echolocation calls and social calls" submitted to Proceedings B.

attract mates over distances [21]. However, amplitude differences in social calls do not code vital information, such as target distance, size, and strength [22, 23].

COMMENT: 3. (lines 68-71) Minor point. This is an interesting idea, but I encourage the authors to consider editing this sentence so it does not presuppose that “bats do not need to perceive the echoes of their own social vocalizations.” I am unaware of evidence that bats use/do not use echoes of social vocalizations to obtain information (but I also know that bats never cease to amaze bat researchers).

RESPONSE: We thank the referee for this insightful comment and have removed this sentence from the introduction.

COMMENT: 4. (lines 83-85) A non-mutually exclusive alternative possibility to why bats have low-frequency hearing (in addition to listening for prey-generated sounds or pup isolation calls) is that sensitivity to low frequencies was crucial and selected for in the earliest mammals for general alertness and detecting predator-generated sounds (e.g. see Fig. 1 in Fullard 1988 *Experientia* 44, 423-428). Perhaps this is what was meant by saying this “may be a remnant from the bat’s phylogenetic ancestors” but the word remnant is tricky because it can suggest that an ability is a relic rather than still functionally useful and adaptive.

RESPONSE: We whole-heartedly agree with the referee in this point and have reworded the section to make the meaning clearer.

Lines 76-84: This low-frequency sensitivity conceivably allows bats to listen to prey- and predator-generated sounds and may have been retained from the bats’ phylogenetic ancestors [3]. Another possibility is that the low-frequency sensitivity evolved in correlation with the frequency content of vital social calls, e.g. pup isolation calls, which are fitness relevant social signals used by dependent offspring to solicit care. Correspondingly, one study detected an evolutionary relationship between bat hearing thresholds and the frequency range of a species’ echolocation and isolation calls [12]. At present, however, it is unresolved whether bats’ low frequency hearing capacities are mainly influenced by the need to detect prey/predator-generated sounds and/or social vocalizations.

METHODS

COMMENT: 5. (lines 101-109) All of the data were collected in Panama but the ms does not say where... indoors in a lab/building or in the field (i.e. more details please)? The ms proper has more information and space devoted to the phylogenetic analysis than the auditory analysis. Because most of the Results rely on the ABR data, I think the methodological information about ABR electrophysiology and recording procedures should be moved into the main body of the ms because it is too important to be relegated to Supplementary Material.

RESPONSE: We agree with the referee that the ABR analysis should be more prominent and we have moved the details about the ABR recording procedure into the main manuscript now. As there are strict text length limitations, we decided to leave the detailed description of the ABR setup and the application of the anaesthetics in the supplementary material. We also added additional information about the catching locations of the wild bats. The ABR measurements were all conducted indoors in the lab space of the Smithsonian Tropical Research Institute in Gamboa, Panama, which is now stated in the manuscript.

Lines 102-105: All animals were adult and wild caught in Panama, near their roosting sites in the area of Gamboa (9.119925, -79.704512), during March and April 2019. All bats were kept for experimentation in the laboratory of the Smithsonian Tropical Research Institute in Gamboa and were released again within 24 hours of their capture.

COMMENT: 6. (lines 112-115) The ms presents clicked-evoked ABRs measured at a high sound pressure level but does not say why click ABRs were collected and their value to the results and data analysis? Figure S3 shows that the mean click-evoked ABR waveforms at 100 dB SPL were quite variable within and between bat species, but there was no further analysis demonstrating whether this variation was meaningful/indicative or predictive of the species. Moreover, there are many sources of

Responses to the two reviews of the manuscript RSPB-2020-1170 entitled "Hearing sensitivity and amplitude coding in bats are differentially shaped by echolocation calls and social calls" submitted to Proceedings B.

variation that could influence the strength and appearance of an ABR waveform (e.g. differences in anesthesia state, electrode placement, skull thickness, electrical noise at time of recording, etc.). Clicked-evoked thresholds were ~60 dB peSPL. Is this because the clicks were shorter in duration and had less energy than the 2.5 ms pure tones? I encourage the authors to add a bit more clarifying text describing the value of collecting and presenting the click-evoked data in the ms given almost all of the ms focuses on the tone-evoked data.

RESPONSE: We agree with referee 2 that the click-evoked ABR data have been neglected in the earlier version of this manuscript. Click-evoked ABRs have historically been recorded in hearing assessment studies. We included these data mainly for the sake of comprehensiveness, but no additional information was gained from these recordings. In order to remove causes for confusion, streamline the content of this manuscript, and focus on the relevant findings, we have now removed the click-evoked ABR data from this manuscript. This does not impact the results of this study in any way and helps to focus on the main results of our study.

COMMENT: 7. (lines 154-157) This analysis assesses the growth in the suprathreshold ABR response with increasing stimulus level but does not compare ABR thresholds within or between a species (except later for a sex difference comparison in four species).

RESPONSE: We agree with the referee that our within- and between-species comparison could be more explicit. However, the within species comparison has been conducted by producing the mean and SEM of the absolute threshold for each species, which indicates the species variation around the mean. We have now expanded on the differences in SEM in the different species in the discussion.

Lines 323-330: In all species we measured, the ABR thresholds showed a general tub shape, even though species differed in the frequency of their respective sensitivity peaks. Our findings show that hearing thresholds are characteristic despite the existing overlap between species (Fig. S3). The small variation between hearing thresholds within species (shown by the small SEM in Fig. 1) and the comparably large variation between species (Fig. S3) indicates species-specific adaptation to their ecological or evolutionary niches. For example, D. rotundus shows an unusually high hearing sensitivity in low frequency ranges (<10 kHz; Fig. 1f), which was previously shown to support prey-generated-noise detection in these sanguivorous bats [44].

Another within species comparison we conducted is the ABR growth function slope analysis. The slope analysis presents an indirect description of the isoreponse lines within one species. The presentation of the ABR growth function slopes (Fig. 3) and the absolute thresholds (incl. SEM, Fig. 1) is a first step towards the between-species comparison. In the result section "Hearing thresholds in relation to echolocation calls and social vocalizations" (line 249) and the discussion ("Amplitude coding in bats follows a general principle" (line 337) and "ABRs are useful tools for cross-species analyses of hearing capacities" (line 321)) the between-species comparison is addressed. In the revised version of the manuscript, we now expanded on this comparison by including an additional summary figure in the electronic supplementary material (new Fig. S3), which shows all measured audiograms in one graphic, thus directly illustrating between-species differences.

Page 6, ESM (caption Figure S3): Comparative summary figure illustrating all eleven species-specific mean hearing thresholds. Colours indicate the different species and line styles indicate family (finely dotted lines: Emballonuridae; solid lines: Phyllostomidae; widely dotted lines: Mormoopidae, Thyropteridae, Vespertilionidae, and Molossidae). Despite the species-specific shape of the thresholds, a general trend of sensitivity increase between 5-15 kHz and decrease above 50-60 kHz is noticeable.

COMMENT: 8. (lines 157-162) I was a bit confused by the description of the slope calculation. I assume that the plotted functions were the amplitudes of the evoked ABR signals at each SPL, and that the slope has units of $\mu\text{V}/\text{dB}$ (the units are missing in Fig. 2). Later (lines 160-161), the text suggests the dynamic range at a given frequency is the ratio of the loudest and faintest stimuli that can be

Responses to the two reviews of the manuscript RSPB-2020-1170 entitled "Hearing sensitivity and amplitude coding in bats are differentially shaped by echolocation calls and social calls" submitted to Proceedings B.

detected. It would be helpful and less confusing to the reader if the definitions and calculations of the slope and the dynamic range of hearing were explicit.

RESPONSE: We thank the referee for this helpful comment. The slope calculations were indeed done as described by the referee and have the unit $\mu\text{V}/\text{dB}$. We have added this unit to Figure 2 and have clarified the definition of the dynamic range in the revised version of the manuscript. We also explain the calculation of the ABR growth function slope in more detail now.

Lines 171-178: To calculate the slope of growth functions at each stimulus frequency, a a sigmoidal curve was fitted to each function (using the nlinfit function in Matlab) and the coefficient of determination (R^2) of the fits, indicating the goodness, was assessed. A shallow slope suggests a large dynamic range, i.e. stimulus level differences are finely coded at this frequency, while a steep slope indicates a smaller dynamic range. Using ABR growth functions to assess the dynamic range of hearing, i.e. the ratio between the loudest and faintest stimuli that can be detected, has mainly been explored in relation to human loudness perception in the past [28, 37, 38], but has also been applied in animal models [26, 27].

RESULTS

COMMENT: 9. (lines 230-240) The Abstract says "ABR thresholds differed between species but showed a general tub-shaped trend" ... and later (lines 25-26) ... "In most species, auditory sensitivity was equally good at both high and low frequency ranges." I mention this text from the Abstract here because the Results section did not compare ABR thresholds between the low and high frequency ranges within and/or between species. There was a comparison of ABR growth functions between the low (pup isolation calls) and high (echolocation calls) frequency ranges, and between males and females in 4 species, but not a comparison of thresholds. I'm not suggesting that the authors conduct a threshold comparison, but they should clarify the text of the Abstract and Results to make it clear that a comparative analysis of threshold tuning was not conducted.

RESPONSE: We agree with the referee that the threshold comparison within and between species has been limited in the previous version of the manuscript. We have now extended the abstract and the comparison in the discussion and result section (for details see response to comment #7). We have also included an additional figure in the supplements (new Fig. S3), directly comparing the audiograms between species.

Lines 20-24 (abstract): We tested if hearing sensitivity and stimulus level coding in bats differ between high and low frequency ranges by measuring auditory brainstem responses (ABRs) of 86 bats belonging to 11 species. In most species, auditory sensitivity was equally good at both high and low frequency ranges, while amplitude was more finely coded for higher frequency ranges.

COMMENT: 10. (lines 235-236) Minor point. To help your readers, please specify the audiograms in Fig. 1 that have three (3) sensitivity peaks and whether they exist in the threshold tuning curves or in the bootstrapped iso-response lines.

RESPONSE: We have clarified, which example we were referring to in the text and have detailed, which curve precisely we were referring to.

Lines 222-226: If three peaks were present, we used the peaks in the lowest and highest frequency regions. This was the case for three species in this study (namely Rhynchonycteris naso (Fig. 1c), Glossophaga soricina (Fig. 1d), and Thyroptera tricolor (Fig. 1i)). We determined the frequency of each sensitivity peak by using either the $1\mu\text{V}$ isoresponse line (own data) or the audiogram threshold line (data from the literature; Table S2).

We also want to point out that the isoresponse lines are not, in contrast to the threshold curves, bootstrapped, i.e. there was no statistical criterion applied to them. We explain this difference in more detail in the methods section of the revised manuscript.

Lines 168-170: The isoresponse lines are not bootstrapped, but instead indicate the species-specific average ABR signal strength (in μV) for each measured frequency-amplitude combination. Isoresponse lines are independent from previous values (e.g. for different amplitudes at the same tested frequency).

Responses to the two reviews of the manuscript RSPB-2020-1170 entitled "Hearing sensitivity and amplitude coding in bats are differentially shaped by echolocation calls and social calls" submitted to Proceedings B.

COMMENT: 11. (lines 252-254) The ms here could be clarified so that readers do not think the distance between iso-response lines is the ABR growth function. Also, please see point #8 on clarifying the calculation and slope units. I was confused by the word "flexible" to describe the two classes of ABR growth functions. How are ABR growth functions with shallower slopes more "flexible" than ABR growth functions with steeper slopes? My understanding is that growth functions with shallower slopes can encode changes in stimulus SPL more finely compared to growth functions with steeper slopes.

RESPONSE: We understand that the calculation of the ABR growth function has been convoluted and not easy to follow in the previous version of the manuscript. We have now clarified the calculation procedure and the meaning of the ABR growth functions in more detail. This explanation should clarify now that they are indeed correlated to the distance of the isoresponse lines.

The referee rightfully points out that "flexible" to describe the growth function is an unfortunate choice of words. The slopes are either "shallow" or "steep". The correlation of the ABR growth functions to the dynamic range of hearing is the reason for our previous use of the word "flexible". We have changed this wording now and detailed the relationship between the ABR growth functions and the dynamic range. We hope that the correct assessment of referee 2, that the growth functions are related to the resolution of the stimulus level encoding, is more explicit in the revised version of the manuscript.

Lines 171-178: To calculate the slope of growth functions at each stimulus frequency, a sigmoidal curve was fitted to each function (using the nlinfit function in Matlab) and the coefficient of determination (R^2) of the fits, indicating the goodness, was assessed. A shallow slope suggests a large dynamic range, i.e. stimulus level differences are finely coded at this frequency, while a steep slope indicates a smaller dynamic range. Using ABR growth functions to assess the dynamic range of hearing, i.e. the ratio between the loudest and faintest stimuli that can be detected, has mainly been explored in relation to human loudness perception in the past [28, 37, 38], but has also been applied in animal models [26, 27].

Lines 272-275: The distances between the isoresponse lines can be expressed as ABR growth functions for each frequency (Fig. 2). The shallower the slope of the growth function, the larger is the dynamic range of ABRs, an indication of more detailed coding of the bat's response to different stimulus levels at a given frequency.

DISCUSSION

COMMENT: 12. (lines 300-304) I'm not sure I understand the point that is being made. Behavioral testing often uses a successive approximation procedure to measure thresholds, but I don't understand why measuring ABRs is better for consistently testing responses to the same stimuli compared to using behavior to consistently test responses to the same stimuli? For example, both techniques can be consistently applied, both have multiple sources of variation, both can adapt/habituate, and both can fail. Perhaps I have misunderstood what is trying to be communicated?

RESPONSE: The referee is correct in their assessment that behavioural testing and ABRs can both be used for consistently testing hearing in animals. Our wording here was ambiguous. We tried to highlight the advantages of ABRs as a very fast method to test large sample sizes (as it does not require training). Behavioural hearing assessments are more time consuming and depend on the training and intrinsic motivation of the animals, but nevertheless a very valuable and important measurement technique. We have now better highlighted the benefits of ABRs for the measurement of hearing thresholds.

Lines 331-336: ABRs are a useful tool to assess response strength to a large, consistent, and reproducible parameter space of auditory stimuli in a large number of animals. We argue that this consistency in tested parameters is optimal for a cross-species comparative approach. Moreover, our approach also allowed us to analyze frequency-specific ABR growth functions,

Responses to the two reviews of the manuscript RSPB-2020-1170 entitled "Hearing sensitivity and amplitude coding in bats are differentially shaped by echolocation calls and social calls" submitted to Proceedings B.

which can be used to investigate the dynamic range of hearing and compare sensory capacities of different species.

COMMENT: 13. (lines 351-352) Minor point. The results of Figure 4b were based on one type of social vocalization—pup isolation calls—so I suggest a slightly more conservative rewording of the second clause of this sentence so that it focuses on pup isolation calls that facilitate care and mother-offspring communication.

RESPONSE: We thank the referee for this insightful comment and have reworded the sentence more conservatively.

Lines 374-377: Our analyses showed that auditory perception in bats is not only shaped and constrained by their faculty of echolocation [3, 7], but also by the vocalizations that bats use for social communication. Our results are in accordance with a previous study [12], which also concluded that bat hearing thresholds evolved in correlation with the frequency range of a species' echolocation calls and isolation calls.

COMMENT: 14. (lines 367-369) See point #11 about “flexibly” encoding SPL. What is meant by “auditory sensitivity”? Is this just absolute threshold? Figure 1 suggests that different species may not be equally sensitive in the low and high frequency ranges.

RESPONSE: The term auditory or hearing sensitivity is indeed referring to the species-specific threshold measured via ABRs. We clarified this in the manuscript.

Lines 20-24: We tested if hearing sensitivity and stimulus level coding in bats differ between high and low frequency ranges by measuring auditory brainstem responses (ABRs) of 86 bats belonging to 11 species. In most species, auditory sensitivity was equally good at both high and low frequency ranges, while amplitude was more finely coded for higher frequency ranges.

COMMENT: 15. (Figure 1) This figure is critical, but the size of each plot is too small, and the panels seem overly complex. The numbers on the iso-response contour line are very small for my eyes, as are the red numbers above the x-axes. There is a wealth of color on this plot, including for other types of social vocalizations that were not the focus of the ms (i.e. is it necessary to show the bandwidths of other types of social calls when the ms is about echolocation versus pup isolation calls? Also, it is hard to tell the difference between red and pink at the small size)? I encourage the authors to make Figure 1 a full-page width, with taller and wider panels and larger font sizes.

RESPONSE: We agree that this figure is crucial to the manuscript and have rearranged the panels of the figure in a way that the figure is now only two columns wide, thus giving the figure more space and allowing for a larger print size. Ultimately, the shape and size of the figure is determined by the printing settings of the journal, but we would like to support the referee's suggestion for a large-sized display of this figure.

We think it is important to leave the different call type frequency indicators in the figure to make this information accessible to the reader. It is true that we are focusing mainly on isolation calls in our analysis, but other social calls should be represented as well and might help readers with a different focus to gain additional information from this figure and put other call types in perspective.

COMMENT: 16. (Figure 2) This figure plots the slope of the ABR growth function at each frequency. The units on the y-axis are not reported (i.e. $\mu\text{Vs/dB}$?). Again, each panel is too small. The R2 numbers are nearly impossible to read without magnification. The panels should be taller and wider, so the figure is a full-page width in the journal. The authors might consider standardizing the location of the bars representing the bandwidths of pup isolation calls and echolocation calls across all figures if they thought this would help improve readability.

RESPONSE: As for figure 1, we have now rearranged the panels of the figure in a way that the figure is now only two columns wide. We also thank the referee for pointing out the lack of a y-

Responses to the two reviews of the manuscript RSPB-2020-1170 entitled "Hearing sensitivity and amplitude coding in bats are differentially shaped by echolocation calls and social calls" submitted to Proceedings B.

axis label, which we have now added. The location of the bars indicating the call bandwidth are indeed standardized in their location on the y-axis and need to vary intrinsically along the x-axis.

COMMENT: 17. (Figure 4) This is another very important figure. The caption for the legend says "Different symbols depict ways in which hearing thresholds were determined. SC: subcutaneous [potential] (ABRs), IC: inferior colliculus [potential] (electrophysiological recordings), PC = posterior colliculus [potential] (electrophysiological recordings)." The legend is a bit confusing because it is a mixture of electrophysiological recording techniques and anatomical locations. All three symbols (SC, IC, PC) refer to recording electrophysiological potentials but IC and PC reference to central auditory nuclei. The location of PC is undefined (i.e. does PC refer to the auditory thalamus and/or cortex or the auditory midbrain and/or brainstem?). Were all of the data collected with extracellular recording or were some from intracellular recording? The caption does not mention filled triangles and behavioral thresholds.

RESPONSE: In the revision, we rephrased the figure legend to be more precise.

Lines 573-577: Different symbols depict electrophysiological (SC, IC, PC, and cochlear potential), physiological (change in heart rate) and behavioural (discrimination paradigm) ways in which hearing thresholds were determined. Electrophysiological data was collected from extracellular recordings subdermally (SC) at the brainstem, at the inferior colliculus (IC) and posterior colliculus (PC) in auditory midbrain and at the round window of the cochlea (cochlear potential).

COMMENT: 18. (Figure S2) Please specify if any of the data recorded with the higher frequency resolution technique were used in the final analysis (e.g. to measure ABR thresholds or to obtain ABR growth functions).

RESPONSE: This is an important point and we have clarified in the revised version of the manuscript that the higher frequency measurements were not used in the calculations of the final analysis.

Lines 142-143: These higher resolution measurements were not used in the calculation of the mean hearing thresholds as slightly different frequencies were tested.

References

- Beattie RC (1988). Interaction of click polarity, stimulus level, and repetition rate on the auditory brainstem response. *Scand Audiol.* **17**, 99-109. doi:10.3109/01050398809070698
- Beery AK, Zucker I (2011). Sex bias in neuroscience and biomedical research. *Neurosci Biobehav Rev.* **35**, 565-572. doi:10.1016/j.neubiorev.2010.07.002
- Cebulla M, Lurz H, Shehata-Dieler W (2012). Evaluation of waveform, latency and amplitude values of chirp ABR in newborns. *Int J Pediatr Otorhinolaryngol.* **78**, 631-636. doi:10.1016/j.ijporl.2014.01.020
- Felsenstein J (1985). Phylogenies and the comparative method. *The American Naturalist* **125**(1), 1-15.
- Fischl MJ *et al.* (2019). Urocortin 3 signalling in the auditory brainstem aids recovery of hearing after reversible noise-induced threshold shift. *The Journal of Physiology* **597**, 4341-4355.
- Grinnell AD (1970). Comparative Auditory Neurophysiology of Neotropical Bats Employing Different Echolocation Signals. *Z. vergl. Physiologie* **68**, 117-153.
- Gröger U, Wiegand L (2006). Classification of human breathing sounds by the common vampire bat, *Desmodus rotundus*. *BMC Biology* **4**: 18. doi:10.1186/1741-7007-4-18
- Hansen TF, Pienaar J, Orzack SH (2008). A comparative method for studying adaptation to a randomly evolving environment. *Evolution: International Journal of Organic Evolution* **62**(8), 1965-1977.

- Responses to the two reviews of the manuscript RSPB-2020-1170 entitled "Hearing sensitivity and amplitude coding in bats are differentially shaped by echolocation calls and social calls" submitted to Proceedings B.
- Howe SW, Decker TN (1984). Monaural and binaural auditory brainstem responses in relation to the psychophysical loudness growth function. *J. Acoust. Soc. Am.* **76**, 787-793.
- Huey RB, Garland Jr T, Turelli M (2019). Revisiting a key innovation in evolutionary biology: Felsenstein's "phylogenies and the comparative method". *The American Naturalist* **193**(6), 755-772.
- Keating AW, Henson Jr OW, Henson MM, Lancaster WC, Xie DH (1994). Doppler-shift compensation by the mustached bat: quantitative data. *J. Exp. Biol.* **188**, 115–129.
- Kössel M, Vater M (1985). The cochlear frequency map of the mustache bat, *Pteronotus parnellii*. *J Comp Physiol A* **157**, 687-697.
- Kössl M (1994). Evidence for a mechanical filter in the cochlea of the 'constant frequency' bats, *Rhinolophus rouxi* and *Pteronotus parnellii*. *Hearing Research* **72**, 73-80. doi:10.1016/0378-5955(96)00005-6
- Liberman MC, Liberman LD, Maison SF (2014). Efferent feedback slows cochlear aging. *The Journal of Neuroscience* **34**, 4599-4607.
- Möhrle D *et al.* (2016). Loss of auditory sensitivity from inner hair cell synaptopathy can be centrally compensated in the young but not old brain. *Neurobiology of Aging* **44**, 173-184.
- Serpanos YC, O'Malley H, Gravel JS (1997). The relationship between loudness intensity functions and the click-ABR wave V latency. *Ear Hear.* **18**, 409-419.
- Silva I, Epstein M (2016). Estimating loudness growth from tone-burst evoked responses. *J. Acoust. Soc. Am.* **127**, 3629-3642.
- Simpson AJR, Reiss JD (2013). The Dynamic Range Paradox: A Central Auditory Model of Intensity Change Detection. *PLoS One* **8**, e57497.
- Thiagavel, J., Santana, S. E., & Ratcliffe, J. M. (2017). Body size predicts echolocation call peak frequency better than gape height in vespertilionid bats. *Scientific Reports*, **7**(1), 1-6.
- Toth CA, Parsons S (2013). Is lek breeding rare in bats? *Journal of Zoology* **291**, 3-11. doi:10.1111/jzo.12069
- Wetekam J, Reissig C, Hechavarria JC, Kössl M (2019). Auditory brainstem responses in the bat *Carollia perspicillata*: threshold calculation and relation to audiograms based on otoacoustic emission measurement. *J Comp Physiol A* **206**, 95–101. doi:10.1007/s00359-019-01394-6
- Zheng QY, Johnson KR, Erway LC (1999). Assessment of hearing in 80 inbred strains of mice by ABR threshold analyses. *Hear. Res.* **130**, 94-107.